# Hyperuniformity at the Absorbing State Transition: Perturbative RG for Random Organization

**Xiao Ma[1], Johannes Pausch[1,2] Gunnar Pruessner[2] and Michael E. Cates[1]**

**1** DAMTP, Centre for Mathematical Sciences, University of Cambridge, Wilberforce Road, Cambridge CB3 0WA, United Kingdom
**2** Department of Mathematics, Imperial College, London SW7 2AZ, United Kingdom

## Abstract

Hyperuniformity, in which the static structure factor or density correlator obeys $S(q) \sim q^\varsigma$ with $\varsigma > 0$, emerges at criticality in systems having multiple, symmetry-unrelated, absorbing states. Important examples arise in periodically sheared suspensions and amorphous solids; these lie in the random organisation (RO) universality class, for which analytic results for $\varsigma$ are lacking. Here, using Doi-Peliti field theory for interacting particles and perturbative RG about a Gaussian model, we find $\varsigma = 0^+$ and $\varsigma = 2\epsilon/9 + O(\epsilon^2)$ in dimension $d > d_c = 4$ and $d = 4 - \epsilon$ respectively. Our calculations assume that renormalizability is sustained via a certain pattern of cancellation of strongly divergent terms. These cancellations allow the upper critical dimension to remain $d_c = 4$, as is known to hold for RO, whereas generic perturbations (*e.g.*, those violating particle conservation) would typically flow to a fixed point with $d_c = 6$. The assumed cancellation pattern is closely reminiscent of a long-established one near the tricritical Ising fixed point. (This has $d_c = 3$, although generic perturbations flow instead towards the Wilson-Fisher fixed point with $d_c = 4$.) We show how hyperuniformity in RO emerges from anticorrelation of strongly fluctuating active and passive densities. Our one-loop calculations also yield the remaining RO exponents to order $\epsilon$, surprisingly without recourse to functional RG methods. These exponents coincide as expected with the Conserved Directed Percolation (C-DP) class which also contains the Manna Model and the quenched Edwards-Wilkinson (q-EW) model. Importantly however, our $\varsigma$ exponent differs from one found via a mapping to q-EW. That mapping neglects a conserved noise term in the RO action, which we argue to be dangerously irrelevant. Thus, although other exponents are common to both, the RO and C-DP universality classes have different exponents for hyperuniformity.

# 1   Introduction

Any configuration that a system can enter, but not escape from, is called an absorbing (or inactive) state. A continuous 'absorbing-state transition' arises when the probability of surviving indefinitely, for a system in the thermodynamic limit of infinite size, falls continuously to zero upon varying a control parameter such as particle density [1, 2].

    Of particular interest are cases where there are many distinct absorbing states that are unrelated by symmetry. For example, experiments on non-Brownian particles of number density $\rho$ supended in a fluid, subject to slow periodic shearing of fixed amplitude, show that at large $\rho$ particles collide and are randomly displaced each cycle: there is always a finite density, $\rho_A$, of 'active' particles. In contrast, below a critical density, $\rho = \rho_c$, particles 'randomly organize' into a non-colliding, stroboscopically static, disordered state in which all particles are passive: $\rho \equiv \rho_A + \rho_P = \rho_P$ [3–5]. At the critical density, this state of random organization (RO)

takes infinitely long to appear, and acquires infinitely long-range correlations as is generic at a second order phase transition. Approaching from within the active phase ($\rho \to \rho_c^+$), the correlation length and time diverge continuously with universal critical exponents [1–6].

Remarkably, the emergent spatial correlations are 'hyperuniform', a term meaning that long-range density fluctuations are completely supressed [7]. Thus in RO, avoidance of collisions requires emergence of a highly correlated configuration in which density fluctuations *vanish* at low wavenumbers $q$, rather than *diverge* as in equilibrium criticality. In dimensions $d = 2, 3$, the static structure factor at criticality vanishes as a power law: $\langle \rho_\mathbf{q} \rho_{-\mathbf{q}} \rangle \equiv \delta^d(0) S(q) \sim q^\varsigma$ with $\varsigma > 0$ [8,9]. Scaling arguments then give $S(0) \sim \xi^{-\varsigma}$ at large but finite correlation length $\xi$. These phenomena are not limited to sheared colloids, but are generic for systems in which a nonconserved, diffusive scalar order parameter ($\rho_A$), is coupled locally to a conserved density ($\rho$), such that there are multiple absorbing states ($\rho_A = 0$) of different frozen density patterns $\rho_P(x)$. This scenario defines the Random Organisation, or RO, universality class.

Until now, the RO class has been assumed identical to a larger one that also includes several different-looking models, including the Manna model of sandpiles [10–12], conserved directed percolation (C-DP) [1,2], and the quenched Edwards-Wilkinson model (q-EW) of interfacial growth. The q-EW correspondence uses a mapping [13–15] in which the interfacial height is the time-integrated active particle density, $u(t) = \int_0^t \rho_A(s)ds$, while the random field evaluated at height $u$ gives $\rho_P$. (Both freeze at the depinning transition.) Functional renormalisation group (FRG) methods applied to q-EW [16] have allowed calculation to order $\epsilon = 4 - d$ of the C-DP exponents $\beta = 1 - \epsilon/9$, $\nu_\perp = \frac{1}{2} + \epsilon/12$ and $z = 2 - 2\epsilon/9$ [14,15], describing the vanishing of the order parameter ($\rho_A \sim (\rho - \rho_c)^\beta$), and the divergences of the correlation length ($\xi \sim (\rho - \rho_c)^{-\nu_\perp}$) and time ($\mathcal{T} \sim \xi^z$). Since the passive particle density field $\rho_P$ is mapped to the random quenched field and integrated out of the q-EW action, it becomes difficult to interpret either the total density field $\rho$ or hyperuniformity. Nevertheless, arguing that the evolution of the total density field is subject to the diffusion of active particles only and not to any dynamical noise, a scaling analysis between $\rho$ and $\rho_A$ yields the hyperuniformity exponent $\varsigma = 0 + \epsilon/3$ [17]. (This calculation of [17] was in part motivated by a pre-publication version of the present work [18].)

In this paper, we find $\varsigma$ to order $\epsilon$ for RO by instead using a perturbative RG (not FRG) for a Doi-Peliti field theory of interacting particles [19–21]. Our hyperuniformity exponent, $\varsigma = 0 + 2\epsilon/9$, differs from the one obtained via the q-EW scaling analysis in [17]. This discrepancy is not a contradiction, but linked to the emergence of two distinct physical scenarios corresponding to the presence (our result) or absence (q-EW result) of *diffusive conservative noise* in the dynamics of the active particles [6, 15, 22]. The two different calculations of $\varsigma$ can both be correct if the conservative noise is *dangerously irrelevant*, and therefore capable of changing the exponents for correlation functions while leaving the remaining exponents intact [23]. In effect, the diffusive noise splits the RO/C-DP/q-EW universality class into two sub-classes for the purposes of studying hyperuniformity. We shall give compelling arguments that exactly this scenario does arise in RO.

Our calculations have several novel technical features which go beyond those normally encountered in perturbative RG calculations using Doi-Peliti field theories. Indeed, prior to the current work, those obstacles have prevented a perturbative field-theoretic calculation not just of $\varsigma$ but of the standard RO/C-DP/q-EW exponents ($\beta, \nu_\perp, z$). Below we accomplish the calculations to one loop, subject to a set of clearly stated assumptions that are needed to resolve various ambiguities that arise.

As well as finding $\varsigma$, we recover to order $\epsilon$ the known values of $\beta, \nu_\perp, z$. This offers a powerful check on our assumptions: it would be a remarkable coincidence for all three exponents to emerge from a perturbative theory that was not well-founded. One may wonder how we can find these results perturbatively, when others have argued that FRG is required to deal

with an infinite number of relevant operators [16]. Yet, because the FRG is performed on the opposite side of the C-DP/q-EW mapping, which involves a non-trivial infinite summation of individual Doi-Peliti operators, this is not necessarily a contradiction.

En route to our perturbative RG results, we examine our interacting particle theory at Gaussian level, applicable for $d > d_c = 4$. (Even this requires a careful treatment of tree-level diagrams.) Surprisingly, we also find a type of hyperuniformity here, albeit of a singular form that can be viewed as an exponent of $\varsigma = 0^+$. This resolves uncertainty [24, 25] over whether hyperuniformity persists in high dimensions where the Gaussian theory should hold. The Gaussian theory lays bare a significant feature of RO (hinted at in [8]): the fluctuations of $\rho_A$ and $\rho_P$ are not separately hyperuniform even as those of $\rho = \rho_A + \rho_P$ become so. This requires near-perfect anticorrelation between the two particle types, which our Gaussian results expose, and our RG results further illuminate. Notably again, we find that the conservative noise plays a central role in the hyperuniformity found at Gaussian level. Indeed, we will see that omitting this noise creates a conservation law on the centre of mass of the particle density distribution, which is known to have drastic effects, including the occurrence of hyperuniformity throughout the active phase rather than just at the critical point [26].

Our calculations therefore (i) unveil the true character of the RO transition, with hyperuniformity emerging from cancellation of large active and passive fluctuations; (ii) directly compute the hyperuniformity exponent as $\varsigma = 0^+$ for $d > 4$ and $\varsigma = 2\epsilon/9 + O(\epsilon^2)$ for $d < 4$, as well as recovering the known exponents to this order without recourse to FRG methods; and (iii) show that conservative noise is required to fully understand the RO universality class whose hyperuniformity exponent is governed by this dangerously irrelevant term in the action.

This paper is structured as follows. In Section 2, we start by discussing some of the technical difficulties in performing perturbative RG for the RO universality class, and enumerate the arguments and assumptions needed to steer a path through these difficulties. Sections 3 and 4 show calculations for critical exponents at Gaussian and one-loop level respectively, and Section 5 discusses several physical aspects of our results. We conclude briefly in Section 6.

## 2   Technical Challenges and Assumptions

Our RG analysis for RO broadly follows procedures for field-theoretic RG [19, 27]. However, our approach is not completely standard but instead presents several technical challenges. Before moving on to the actual calculations in Sections 3 and 4, we discuss these challenges in Section 2.1. To resolve them, we make several assumptions. Briefly these are (i) universality of the RO model, subject to the role of the dangerously irrelevant noise, (ii) perturbative renormalisability of the theory in Doi-Peliti formalism and (iii) emerging hyperuniformity (rather than divergent fluctuations) at the critical transition. Reasoned arguments for these assumptions, and more precise statements of them, are given in Section 2.2. Both Sections 2.1 and 2.2 are somewhat technical and might be skipped on a first reading of the paper, especially by non-specialists. However, we feel it is much clearer to lay out these assumptions coherently in advance, rather than introducing them piecemeal at various points during the calculations.

### 2.1   Overview of Technical Difficulties

The first technical challenge is the interpretation of the Doi-Peliti action, whose final (shifted) form is given in Eq. 28 below. The Doi-Peliti field formalism involves a coherent state path integral directly built from the master equation, where the primary fields of the theory are not direct particle densities but conjugate creation and annihilation operators. For this reason, physical observables such as the two-point correlation functions of active and passive particles are typically sums of terms that may scale differently [28]. Based on a Fokker-Planck equation

[29], the Doi-Peliti formalism incorporates diffusion and diffusive noise differently compared to the response field formalism based on a Langevin equation of the particle densities. While the former contains a single squared gradient term representing the (stochastic) diffusion, the latter introduces a deterministic squared gradient as well as a conserved noise, which is multiplicative in general [30]. We will argue later that this diffusive noise is dangerously irrelevant.

Both in understanding the physical noise implemented by the Doi-Peliti action, and in calculating the observables, problems arise if these are interpreted from a response-field perspective that does not respect the nature of the fields [22, 31] or ignores their commutator relations [32]. However, as long as the field-theoretic RG machinery in the Doi-Peliti formalism is strictly followed, perturbative RG on it is known to be capable of producing correct exponents for important universality classes. Examples include directed percolation [1, 19] and branching random walks [21]. Such calculations are unambiguous either because there are only a few relevant couplings throughout the RG calculations, or because some type of Ward identity links couplings into a small number of combinations which then flow in lockstep. However, the third of our technical challenges is that more generally (including for RO), there is a plethora of relevant couplings, present in the bare action or potentially generated at the fixed point, which might in principle contribute. To study the RO universality class, extreme care is then needed to preserve the implicit symmetries between couplings (including but not limited to conservation of total particle number) so that the fixed point calculated through the RG analysis is indeed that of RO, rather than some other two-species reaction-diffusion model with, typically, an upper critical dimension of 6 instead of 4 [33, 34].

An additional complication, closely related to the one just outlined, is that there are more relevant parameters in the bare theory than the final effective one. This means that the concept of an RG fixed point has to be replaced by that of an RG fixed-point manifold (FPM). Any point on this manifold is a fixed point of the RG flow functions. This differs from the critical manifold, which is a higher dimensional manifold of starting parameters that flow under RG to the RO FPM (in the Wilsonian RG sense). Choosing a judicious form of the fixed point action can ease the perturbative RG calculations considerably and we exploit this in our approach below. We discuss the FPM concept further in Appendix A.

Another feature the RO theory is that since passive particles do not diffuse, the passive propagator is momentum-independent. Together with certain generated vertices absent in the bare action, this leads to loop diagrams that exhibit non-renormalisable divergences both in the IR and UV regime. Examples of such super-divergent loops will be discussed later and are presented in Appendix B.

Non-renormalisable divergences in the IR regime can arise when there is one or more unstable direction leading away from the RO fixed point (with $d_c = 4$) to a generic reaction-diffusion one (with $d_c = 6$). The role of these divergences can be understood by looking at the tricritical fixed point in the Ising model [35]. There, working near the tricritical fixed point, one encounters divergences that would be individually non-renormalisable and are inversely proportional to the mass (distance away from criticality). These are caused by an unstable direction leading from the tricritical fixed point to the critical (Wilson-Fisher) one which has higher upper critical dimension ($d_c = 4$ rather than $d_c = 3$ for the tricritical point). They cancel when the tricritical fixed point is approached along a path that has no component in the unstable direction and therefore does actually arrive at the fixed point one is trying to study [35]. We will assume below that the same happens in RO.

Non-renormalisable UV divergences arguably pose a bigger and more serious problem. The most rigorous method to resolve such divergence issues is to explicitly show that all non-renormalisable divergences across all relevant couplings cancel, for example by constantly observing certain Ward identities/symmetries between the large number of couplings in order

to stay in the RO class. While proving this cancellation explicitly is beyond the scope of the present work, we shall assume that RO cast into a Doi-Peliti field theory is perturbatively renormalisable. A direct consequence of this assumption is that cancellations have to occur for all couplings of this non-renormalisable "problematic" type that may be generated at the fixed point. Therefore, assuming renormalisability of our theory is enough to ensure that all the problematic couplings, whose bare values are zero in RO, must remain suppressed via cancellations both in the judicious form of the fixed point action and throughout the RG calculations.

So far, exploiting the degeneracy provided by the FPM and the perturbative renormalisability assumption, we have argued that a sensible starting point action of the RG calculations exists. However, there remains a final risk concerning the treatment of vertices that may be generated at the fixed point but not present in the bare theory, which can be crucial to finding the correct exponents. One example of such a scenario is the Fredrickson-Andersen (FA) model [36], where a linear diffusion term is absent in the microscopic description. Though only generated during the flow, the theory without a diffusion term is fundamentally incomplete even at Gaussian level and hence in $d > d_c$. The generated relevant coupling therefore has to be manually put into the starting point action at $O(\epsilon^0)$; without it, the Gaussian starting point is not within $O(\epsilon)$ proximity of the final effective action of the nonlinear model in $d < d_c$, and the resulting theory certainly fails to be perturbatively renormalisable. Correspondingly, the diffusive coupling in FA, once added to the bare theory, must be treated just like the $O(1)$ couplings already present there, so it changes the values of loops used to calculate exponents at $O(\epsilon)$. In making such manual modifications of the action, however, one has to be mindful to not break implicit symmetries, as this could lead to a theory outside the intended universality class. Indeed this was shown for FA [37], which explained why earlier work on that model appeared to give $d_c = 4$ when in fact $d_c = 2$. For this reason, in problems of this type, it is not appropriate to add all relevant terms in the initial action, even if not present already there, purely as a precaution against these being generated at the fixed point. Specifically, one should not add couplings that are non-renormalisable near $d_c = 4$ because these generically perturb the RO theory into one with $d_c > 4$.

We can summarize our approach to the apparent divergences as follows. At the RG fixed point of the RO theory, certain problematic couplings (such as the ones exhibited in Appendix B) may be generated, producing loop diagrams with non-renormalisable IR and UV divergences. We assume that the Gaussian theory without these couplings makes sense *and* that the nonlinear theory is renormalisable, indicating that these couplings must cancel and can safely be suppressed. A more precise statement of this and other assumptions follows below.

## 2.2 Statement of assumptions used for the RG calculations

In this Section, we gather the various arguments and assumptions needed to overcome the technical difficulties surveyed above, so that our perturbative RG calculations can give unambiguous exponent predictions at order $\epsilon$.

1. *Universality*

   (a) We consider the Random Organization (RO) model where a non-conserved order parameter ($\rho_A$) is coupled to a conserved density ($\rho$), and can undergo absorbing state phase transitions into infinitely many absorbing state configurations. We **argue** that this describes a large universality class, which we hereafter refer to as the RO class. A minimal realization is a reaction-diffusion system consisting of an 'active' diffusing species (diffusivity $D$) and a 'passive' non-diffusing species, with number-conserving reactions $A \to P$ and $A + P \to A + A$. The intrinsic noise in

the Doi-Peliti theory, when written in terms of Langevin equations, include both a birth-and-death, multiplicative noise $\sim \sqrt{\rho_A}\eta$, and a diffusive, conserved noise $\sim \sqrt{D\rho_A}\nabla \cdot \Xi$. Here $\eta$ and $\Xi$ are Gaussian white spatiotemporal noises of suitable dimensionality.

(b) The RO class we propose differs from the customary C-DP/Manna/q-EW class by including the diffusive conserved noise. Since this term is RG-irrelevant, we **argue** that the RO universality class has the same upper critical dimension $d_c = 4$ as C-DP/Manna/q-EW, and shares the same order parameter exponent $\beta$, dynamic exponent $z$, and correlation length exponent $\nu_\perp$.

(c) We **argue** that the diffusive noise, despite its RG-irrelevance, can alter the hyperuniformity exponent describing total density correlation functions, because it is *dangerously* irrelevant. This can create an important distinction between RO and C-DP/Manna, while leaving $\beta, z, \nu_\perp$ and $d_c$ equal for the two classes. We elaborate this reasoning further in Sections 3.3 and 5.2 below. In particular we **define** the RO class to exclude cases where the particles move only by centre-of-mass-conserving interactions, such as those considered in [26], whose additional conservation law eliminates the diffusive noise.

(d) We **argue** that, because of the presence of more relevant parameters in the bare theory than in the final effective one, the concept of an RG fixed point has to be replaced by that of an RG fixed-point manifold (FPM); see Appendix A for details. We **assume** that any microscopic theory that flows to the shared FPM under RG is a member of the RO universality class.

2. *Renormalisability*

(a) We **assume** that the Doi-Peliti theory for RO is renormalisable around its upper critical dimension $d_c = 4$.

(b) Given this and 1d above, we **argue** that starting at a point on the Gaussian FPM, in $d = 4-\epsilon$ the renormalised nonlinearities corresponding to a microscopic theory in the RO class will generically correspond to a fixed point somewhere on RO's FPM in the IR limit, which we **assume** is in $\mathcal{O}(\epsilon)$ vicinity of the Gaussian FPM. This allows us to study the fixed point governed by a particular microscopic action rather than one containing all possible relevant terms.

(c) Within a general reaction-diffusion setting there are many nonlinearities that violate the main precept of RO and/or C-DP classes (namely, an infinite number of absorbing states that are not symmetry-related). For the generality of these models $d_c$ is not 4 but 6. Such terms enter our initial action for RO in specific combinations. Yet taken individually, each can generate problematic, algebraic (not logarithmic) divergences in $d = 4$ which we **argue** (given that $d_c = 4$, see assumption 1b above) must cancel.

(d) We **assume** that this cancellation follows a broadly similar scenario to that for the equilibrium tricritical Ising model ($d_c = 3$), at least in the IR regime.

3. *Hyperuniformity*

We **assume** that the spatial correlations in total density are hyperuniform at criticality in RO. This assumption is supported by simulation data [8,9]. It not necessary for determining $z$, $\nu_\perp$, or any other critical exponents that can be found via scaling relations from these two. It is however needed to extract $\beta$ (to order $\epsilon$), as well as the hyperuniformity exponent $\varsigma$, from our perturbative RG approach. We refer to

Section 5 for a discussion of the numerical evidence and of further implications of this assumption.

## 3 The Gaussian Theory

We now study the RO universality class at Gaussian level, starting from a particular microscopic realization.

### 3.1 Field Theory

We consider a lattice model comprising $A$ particles that hop with diffusivity $D$ and non-hopping $P$ particles. The on-site reaction $A + P \rightarrow 2A$ has rate $\kappa$, causing passive particles to awaken on encounter with active ones; the reaction $A \rightarrow P$ has rate $\mu$ so that active particles decay to passivity without collisions. Following established procedures [19, 38] we start by writing the master equation for the model. Letting $n_i$ denote the number of active particles at position $i$ and $m_j$ the number of passive particles at position $j$, the master equation is

$$\partial_t P(n, m, t) = \mu \sum_i \Big( (n_i + 1) P(n + 1_i, m - 1_i, t) - n_i P(n, m, t) \Big)$$
$$+ \kappa \sum_i \Big( (n_i - 1)(m_i + 1) P(n - 1_i, m + 1_i, t) - n_i m_i P(n, m, t) \Big) \qquad (1)$$
$$+ \frac{D}{h^2} \sum_{\langle i,j \rangle = h} \Big( (n_j + 1) P(n + 1_j - 1_i, m, t) - n_i P(n, m, t) \Big)$$

Here $n$ and $m$ are shorthands for the collections of all $n_i$ and $m_j$ respectively, and where $1_i$ is used to add or subtract a single particle at position $i$; $h$ is the distance between neighbouring sites and $\langle i, j \rangle = h$ sums over all neighbours. We then rewrite the master equation in terms of annihilation operators $\hat{\mathbf{a}}, \hat{\mathbf{p}}$ and creation operators $\hat{\mathbf{a}}^\dagger = \tilde{\mathbf{a}} + 1, \hat{\mathbf{p}}^\dagger = \tilde{\mathbf{p}} + 1$ for $A$ and $P$ particles respectively, where site- and time-indices are suppressed to ease notation. Via a coherent-state path integral and the continuum limit [19], we arrive at a Doi-Peliti action $\mathcal{A} = \int \mathbb{A} \, d^d x \, d t$ with Lagrangian density

$$\mathbb{A} = -\tilde{a}(\partial_t - D\nabla^2)a - \tilde{p}\partial_t p + \mu(\tilde{p}a - \tilde{a}a) + \kappa(\tilde{a}^2 ap + \tilde{a}ap - \tilde{a}a\tilde{p}p - a\tilde{p}p) \qquad (2)$$

in terms of the fields $a(x, t), \tilde{a}(x, t), p(x, t), \tilde{p}(x, t)$, where we have used bold symbols to denote operators and plain symbols for fields, and the arguments of the fields in the above expression have been suppressed for clarity. At mean-field level, we identify the active particle annihilation field that minimises the action with the mean-field density $\rho_A$, and the minimising passive particle annihilation field with $\rho_P$. This gives the expected equations of motion for the global densities at mean-field level, $\dot{\rho}_A = -\mu\rho_A + \kappa\rho_P\rho_A$ and $\dot{\rho} = 0$.

The spatially averaged total density of active and passive particles $\rho = \rho_A(t) + \rho_P(t)$ does not evolve in time, and acts as a control parameter of the system. When $\rho$ is larger than a critical value $\rho_c = \mu/\kappa$, there are two stationary solutions to the mean-field equations: an unstable one where $\rho_A = 0$ (the absorbing state), and the other stable solution with $\rho_A = (\rho - \rho_c)^1 > 0$, $\rho_P = \rho_c$. Thereby the well-established mean-field critical exponent $\beta = 1$ [2] is confirmed. At mean-field level, the system stays in the active phase indefinitely whenever $\rho > \rho_c$. Meanwhile for $\rho < \rho_c$, the only stable solution is $\rho_A = 0$, and the system is in the absorbing phase.

### 3.2 Gaussian Structure Factors: Hyperuniformity via Anticorrelation

Next we expand (2) about the mean field solution and shift $a(x, t) = a_0 + \breve{a}(x, t)$ and $p(x, t) = p_0 + \breve{p}(x, t)$ [20]. The interpretations of these shifts are more than simply perturbations away from mean

densities; rather, $a_0$ and $p_0$ are the mean densities of a set of *Poisson-distributed* active and passive particles *initialised in the distant past* [32]. Using active particles as an example, an $a_0$-Poisson distribution at site $i$ can be initialised at time $t_0$ at the operator level:

$$|\mathcal{M}(t_0)\rangle = e^{-a_0} \sum_{l=0}^{\infty} \frac{a_0^l}{l!} |l_i, 0\rangle = e^{-a_0} \sum_{l=0}^{\infty} \frac{a_0^l \hat{\mathbf{a}}_{\mathbf{i}}^{\dagger l}}{l!} |0, 0\rangle \qquad (3)$$

This initialisation can be applied at every position and in order to shorten the notation, we drop the subscript $i$.

If the expectation of an observable $\mathcal{O}$ is evaluated at time $t_1 > t_0$, the initialisation can be compactly written as an additional term that is added to the action (at the field level):

$$\left\langle \mathcal{O}(t_1) e^{-a_0} \sum_{l=0}^{\infty} \frac{a_0^l a^{\dagger l}(t_0)}{l!} \right\rangle = \int \mathcal{D}[\tilde{a}, a] \mathcal{O}(t_1) e^{\mathcal{A} + a_0(a^{\dagger}(t_0) - 1)}$$

$$= \int \mathcal{D}[\tilde{a}, a] \mathcal{O}(t_1) e^{\mathcal{A} + a_0 \tilde{a}(t_0)} \qquad (4)$$

This additional term can be absorbed by shifting the annihilation field by $a_0$ from time $t_0$ onwards, $a(x, t) = \breve{a}(x, t) + a_0 \Theta(t - t_0)$: in the propagator, $-\tilde{a}(x, t)\partial_t a(x, t)$ is replaced by $-\tilde{a}(x, t)\partial_t \breve{a}(x, t) - a_0 \tilde{a}(x, t)\delta(t - t_0)$. Integrated in time, the second term becomes $-a_0 \tilde{a}(t_0)$, hence cancelling the extra term created by the initialisation in (4). We then push back the initialisation time $t_0 \to -\infty$ which makes the Heaviside function $\Theta(t - t_0)$ obsolete.

The fact that the initialisation is pushed back to $t_0 \to -\infty$ implies that, if left unperturbed, the system is in steady state at any finite time $t$. In this Section we do leave the system unperturbed after the initialisation at $t_0 = -\infty$, and the only observables that we calculate are the correlators for the various particle types.

At Gaussian (linear) level, the Poissonian mean densities $a_0, p_0$ remain unchanged under time evolution; consequently, $a_0 = \rho - \rho_{c,g}$ and $p_0 = \rho_{c,g} = \mu/\kappa$, where $\rho_{c,g}$ now carries the suffix $g$ to indicate that it is the bare Gaussian value of the critical density – which is also the mean field value. In particular, $a_0 = 0$ identifies the critical point at the Gaussian level; crucially, this feature of the Gaussian model will change in Section 4 below for the nonlinear theory. The quadratic part of the Lagrangian of the Doi-Peliti theory is thereby found as

$$\mathbb{A}_{G,DP} = -\tilde{a}(\partial_t - D\nabla^2)\breve{a} - \tilde{p}(\partial_t + \kappa a_0)\breve{p} + \kappa a_0 \tilde{a}\breve{p} + \kappa a_0 p_0(\tilde{a}^2 - \tilde{a}\tilde{p}) \qquad (5)$$

Propagators can be read off from here, but as already discussed in Section 2.1, in Doi-Peliti theory there is a nontrivial relation between terms in the action and the physical densities and noises [19, 38]. This means that calculating static density correlators $S_{\alpha\beta}(q)$ for $\alpha, \beta \in A, P$ requires a tree-level computation.

The particle number operator $\hat{\rho}_{\mathbf{A}} := \hat{\mathbf{a}}^{\dagger}\mathbf{a}$ is translated to $\langle \rho_A(x, t) \rangle = \langle a^{\dagger}(x, t)a(x, t) \rangle$ at the field level. The covariance in spatial Fourier space and temporal direct space between active particle densities is calculated in Doi-Peliti field theory as

$$\langle \rho_A \rho_A' \rangle = \langle a^{\dagger}aa'^{\dagger}a' \rangle = \langle aa'^{\dagger}a' \rangle = \langle a\tilde{a}'a' \rangle + \langle aa' \rangle$$

$$= a_0^2 + a_0 \langle \breve{a} \rangle + a_0 \langle \breve{a}' \rangle + \langle \breve{a}\tilde{a}' \rangle + a_0 \langle \breve{a}\tilde{a}' \rangle + \langle \breve{a}\tilde{a}'\breve{a}' \rangle \qquad (6)$$

where a prime indicates that the field has arguments $(-q, t')$, whereas without the prime the field has arguments $(q, t)$ in spatial Fourier space. From this point onward, we will remain at the field level rather than the operator level, and this shorthand notation for arguments will

be used consistently. Then,

$$
\begin{aligned}
\delta^d(q+q')\text{Cov}[\rho_A, \rho_A'] &= \langle \rho_A \rho_A' \rangle - \langle \rho_A \rangle \langle \rho_A' \rangle \\
&= a_0^2 + a_0 \langle \breve{a} \rangle + a_0 \langle \breve{a}' \rangle + \langle \breve{a}\breve{a}' \rangle + \\
&+ a_0 \langle \breve{a}\tilde{a}' \rangle + \langle \breve{a}\tilde{a}'\breve{a}' \rangle - (a_0 + \langle \breve{a} \rangle)(a_0 + \langle \breve{a}' \rangle) \\
&= a_0 \langle \breve{a}\tilde{a}' \rangle + \langle \breve{a}\breve{a}' \rangle - \langle \breve{a} \rangle \langle \breve{a}' \rangle + \langle \breve{a}\tilde{a}'\breve{a}' \rangle
\end{aligned}
\tag{7}
$$

This expression confirms our statement in Section 2.1 that the correlators involves sums over several scaling fields of the theory, so that to find their scaling one has to see which terms dominate on the right hand side.

At Gaussian level, the third and fourth terms in (7) give zero contributions, since these rely on 'source vertices' (these are couplings $\tilde{a}$, $\tilde{p}$, contributing to $\langle \breve{a} \rangle$, $\langle \breve{p} \rangle$ at tree-level, e.g. $\langle \breve{a} \rangle = \underline{\quad\quad}_\times$ ), which retain values of zero in the Gaussian theory. Up to the prefactor $a_0$, the first term equals the propagator, which can be read off from the quadratic part of the action in Fourier space

$$
\langle \breve{a}(q,t)\tilde{a}(-q,t') \rangle = \delta^d(0) \int \frac{e^{-i\omega(t-t')}}{-i\omega + Dq^2} d\omega
\tag{8}
$$

where the formal factor $\delta^d(0)$ equates to the volume of the system (this factor will also arise in Equations (9)–(16) below). Meanwhile the second contribution $\langle \breve{a}\breve{a}' \rangle$ in (7) can be calculated in Fourier space as the sum of two Feynman diagrams, using standard rules for calculating such observables. The diagrams are

$$
\kappa p_0 a_0 \;\hat{=}\; \delta^d(0) \int \frac{2\kappa p_0 a_0 e^{-i\omega(t-t')}}{(-i\omega + Dq^2)(i\omega + Dq^2)} d\omega
\tag{9}
$$

where the factor of 2 comes from the symmetry factor of the active-active noise vertex, and

$$
-\kappa p_0 a_0 \;\hat{=}\; -\delta^d(0)\kappa p_0 a_0 \int \frac{e^{-i\omega(t-t')}\kappa a_0}{(-i\omega + Dq^2)(i\omega + Dq^2)(i\omega + \kappa a_0)} d\omega d\omega'
\tag{10}
$$

which appears on the right-hand side of (7) once in this form and once with dashed and undashed variables interchanged.

Adding these contributions together and taking the inverse temporal Fourier transform gives the active-active covariance at wavevector $q$ as

$$
\text{Cov}[\rho_A(q,t), \rho_A(-q,t')] = a_0 \left( e^{-Dq^2|t-t'|} + \kappa p_0 \frac{Dq^2 e^{-Dq^2|t-t'|} - \kappa a_0 e^{-\kappa a_0|t-t'|}}{(Dq^2 + \kappa a_0)(Dq^2 - \kappa a_0)} \right)
\tag{11}
$$

Similarly the passive-passive covariance equals

$$
\delta^d(q+q')\text{Cov}[\rho_P, \rho_P'] = p_0 \langle \breve{p}\tilde{p}' \rangle + \langle \breve{p}\breve{p}' \rangle - \langle \breve{p} \rangle \langle \breve{p}' \rangle + \langle \breve{p}\tilde{p}'\breve{p}' \rangle
\tag{12}
$$

Since there is no tree-level diagram that represents a two-point passive-passive noise vertex, contributions like $\langle \breve{p}\breve{p}' \rangle$ on the right hand side are zero. The only contributing part is the first term, which is the passive propagator multiplied by $p_0$; hence we directly obtain

$$
\text{Cov}[\rho_P(q,t), \rho_P(-q,t')] = p_0 e^{-\kappa a_0|t-t'|}
\tag{13}
$$

Finally, the active-passive covariance is calculated as follows

$$\delta^d(q+q')\text{Cov}[\rho_A,\rho_P'] = \Theta(t-t')\big(\langle\breve{a}\tilde{p}'\breve{p}'\rangle + p_0\langle\breve{a}\tilde{p}'\rangle\big) + \Theta(t'-t)\big(\langle\breve{p}'\tilde{a}\breve{a}\rangle + a_0\langle\breve{p}'\tilde{a}\rangle\big) + \langle\breve{a}\tilde{p}'\rangle - \langle\breve{a}\rangle\langle\breve{p}'\rangle$$

(14)

This has contributions from 'transmutation', whose propagator connects the two species

$$\langle\breve{a}(q,t)\tilde{p}(-q,t')\rangle = \delta^d(0)\int \frac{\kappa a_0 e^{-i\omega(t-t')}}{(-i\omega + Dq^2)(-i\omega + \kappa a_0)}d\omega$$

(15)

and from the active-passive noise vertex,

$$-\kappa p_0 a_0 \triangleq \delta^d(0)\kappa p_0 a_0 \int \frac{-e^{-i\omega(t-t')}}{(-i\omega + Dq^2)(i\omega + \kappa a_0)}d\omega$$

(16)

When combined and inverse Fourier transformed in time, these give

$$\text{Cov}[\rho_A(q,t),\rho_P(-q,t')] = \Theta(t-t')\frac{\kappa a_0 p_0}{Dq^2 - \kappa a_0}\left(e^{-\kappa a_0|t-t'|} - e^{-Dq^2|t-t'|}\right)$$
$$-\kappa p_0 a_0 \frac{\Theta(t-t')e^{-Dq^2|t-t'|} + \Theta(t'-t)e^{-\kappa a_0|t-t'|}}{(Dq^2 + \kappa a_0)} \quad (17)$$

Adding all three covariances together, we obtain finally the covariance for the total density $\rho(x,t) = \rho_A(x,t) + \rho_P(x,t)$ as

$$\text{Cov}[\rho(q,t),\rho(-q,t')] = a_0 e^{-Dq^2|t-t'|}\left(1 - \frac{Dq^2\kappa p_0}{(Dq^2 + \kappa a_0)(Dq^2 - \kappa a_0)}\right) + p_0 e^{-\kappa a_0|t-t'|}\left(\frac{(Dq^2)^2}{(Dq^2)^2 - (\kappa a_0)^2}\right)$$

(18)

Taking $t = t'$ gives the equal-time structure factor,

$$S(q) = a_0 + p_0\frac{Dq^2}{Dq^2 + \kappa a_0} = a_0 + p_0\frac{(q\xi)^2}{1 + (q\xi)^2}$$

(19)

where $\xi = \sqrt{D/(\kappa a_0)} \sim (\kappa a_0)^{-1/2}$ is the correlation length. This confirms the mean-field critical exponent $\nu = 1/2$. The critical point is at $a_0 \propto \xi^{-2} \to 0$. This implies vanishing of $S(0)$, and hence hyperuniformity, so long as the $q \to 0$ limit is taken first, whereas for finite $q$, $S(q)$ approaches a constant, $p_0$, as $a_0 \to 0$. Therefore at criticality $S(q)$ is zero at the origin but $p_0$ elsewhere, which can be formally viewed as $S(q) \sim q^\varsigma$ with exponent $\varsigma = 0^+$. Note, however, that this order of limits formally reverses the one used in RG to access the critical scaling.

Fig. 1 shows the three structure factors, and sampled density profiles (in $d = 1$ for simplicity), at small $a_0$. This strikingly demonstrates how hyperuniformity emerges by anticorrelation of passive and active particles. This must be so, because the $S_{AA,PP}(0)$ correlators found above each remain finite at criticality ($a_0 \to 0$), where the full density has $S(0) = 0$. While the Gaussian-level calculation does not enforce separate positivity of the cancelling particle densities $\rho_A$ and $\rho_P$, we show later (Section 5) that doing so requires strong non-Gaussianity only in dimension $d < d_c = 4$.

In the RO theory constructed from the Doi-Peliti formalism, there is no unique fixed point action, with couplings replaced by their renormalised values at the RG fixed point, but a *manifold* of fixed point actions (the FPM). This is since each reaction gives rise to more coupling constants than there are independent *effective* coupling constants, and degeneracy of the 'fixed

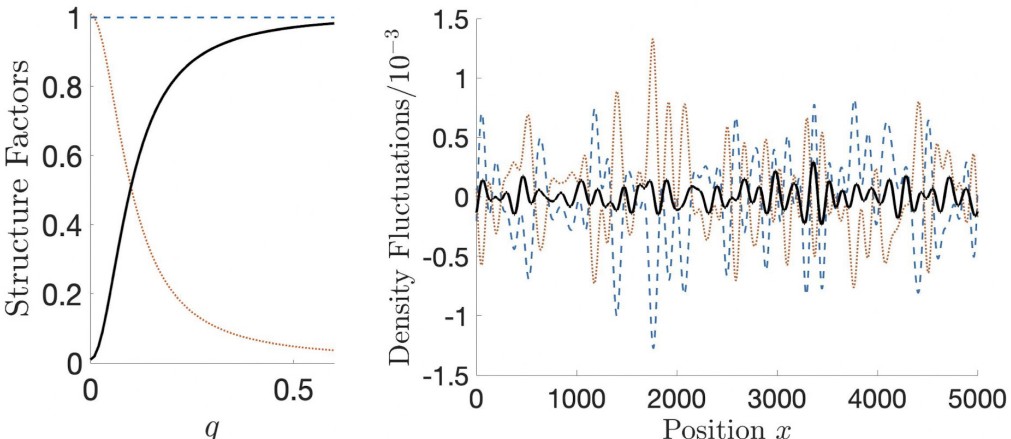

Figure 1: (a) Plot of structure factors $S(q), S_{AA}(q), S_{PP}(q)$ vs $q$ for $a_0 = 0.01, p_0 = \kappa = D = 1$ (giving $\xi = 10$) showing at low $q$ the cancellation-induced suppression of total density fluctuations. Blue dashed line (horizontal) passive; red dotted line (decreasing) active; black solid line (increasing) total density. (b) Sample of spatial density statistics for the Gaussian model in $d = 1$. Parameters as for (a); bold black line is the total density.

point action' arises. While the detailed discussion for the FPM is left to Appendix A, we now briefly show this degeneracy in $d > 4$ dimensions for the Gaussian theory. Here, the perturbative terms (terms not in the harmonic part) of the RO model constructed from any subset of allowed reactions are zero for any fixed point on the Gaussian FPM, and what remain are purely quadratic terms that only follow engineering dimensions. Therefore, the conventionally defined 'fixed point action' of the Gaussian theory is simply the Gaussian part of the original action, with values $\alpha_1, \alpha_2, \alpha_3$ given by the microscopic reactions chosen (examples below):

$$\mathbb{A} = -\tilde{a}(\partial_t - D\nabla^2)\breve{a} - \tilde{p}(\partial_t + \tau_p)\breve{p} + \tau_p \tilde{a}\breve{p} + \alpha_1 \tilde{a}^2 + \alpha_2 \tilde{a}\tilde{p} + \alpha_3 \tilde{p}^2 \tag{20}$$

- For reactions $\{A \to P, A + P \to A + A\}$, $\alpha_3 = 0$ in the Gaussian fixed point action.

- For reactions $\{A \to P, A + P \to A + A, A + A + P \to A + P + P\}$, all three $\alpha$ terms are present in the Gaussian fixed point action. Note that although RG-relevant, the last of these reactions does not change the universality class.

The resulting Gaussian exponents do not differ, despite being calculated from different Gaussian fixed point actions. For example, the total density correlator obeys

$$\text{Cov}[\rho(q, t)\rho(-q, t)] = a_0 + p_0 - \frac{\alpha_1}{Dq^2 + \tau_p} + \frac{\alpha_3}{\tau_p} + (\alpha_1 + \alpha_2 + \alpha_3)\frac{1}{Dq^2} = \frac{\alpha_1}{Dq^2 + \tau_p} + \frac{\alpha_3}{\tau_p} \tag{21}$$

where in the last step we have exploited the symmetry $\alpha_1 + \alpha_2 + \alpha_3 = 0$ specific to the RO class; this will be proved in Section 4.3. In particular, this symmetry is exactly what ensures the Gaussian RO fixed point to exhibit hyperuniform, not diverging, fluctuations in total density. This exemplifies for the Gaussian case how a degeneracy of the fixed point action arises (here between the values of $\alpha_{1,2,3}$). Importantly, for $d > d_c = 4$, not all relevant parameters have to be present in the initial action for RG to give the correct exponents, as long as the one starts on the correct (Gaussian RO) fixed-point manifold.

### 3.3 Hyperuniformity and Conservative Noise at Gaussian Level

A further aspect of the Gaussian theory is exposed by using the Cole-Hopf transformation [38] to find the equivalent Langevin equations, from which the same structure factors as found above can alternatively be derived.

In response field formalisms, annihilation fields in the action represent real density fields, conjugated with artificial creation fields. Averaging over the white noise in the Langevin equations through completing the square shows that quadratic terms in the creation fields represent the statistical weight of the action (details of this derivation can be found in Chapter 4.1 in [19]). However, naively applying the same derivation to Doi-Peliti often yields imaginary noise in reaction-diffusion systems, including RO [15]. This is because, in Doi-Peliti field theory, the annihilator field is the complex conjugate of the creator field, whereas they are independent in the response field formalism. Only in the latter, can anything be integrated out at all without carrying out the whole path integral.

This problem is overcome by a Cole-Hopf transformation [38], creating a response-field dual of the Doi-Peliti action, in order to obtain an action in terms of active and passive particle density fields, $\alpha$ and $\pi$, and their response fields $\tilde{\alpha}$ and $\tilde{\pi}$. Here

$$a = \exp(-\tilde{\alpha})\alpha \qquad a^\dagger = \exp(\tilde{\alpha}) \tag{22a}$$

$$p = \exp(-\tilde{\pi})\pi \qquad p^\dagger = \exp(\tilde{\pi}) \tag{22b}$$

Notice that the dual annihilation field obeys $\alpha = a^\dagger a$ and so represents a real density. Again performing a shift $\alpha \to \alpha + \alpha_0$ and $\pi \to \pi + \pi_0$ and keeping only the quadratic parts, the Cole-Hopf action for RO is at Gaussian level

$$\mathbb{A}_{\text{G,CH}} = -\tilde{\alpha}\partial_t\alpha + D(-\nabla\tilde{\alpha}\nabla\alpha + (\nabla\tilde{\alpha}^2)\alpha_0) - \tilde{\pi}\partial_t\pi + \kappa(\tilde{\alpha} - \tilde{\pi})\alpha_0\pi + \mu(\tilde{\alpha} - \tilde{\pi})^2\alpha_0 \tag{23}$$

There are two advantages of this Cole-Hopf action. Firstly, since $\alpha$ and $\pi$ are particle density fields, the active density correlator can be written neatly as $\langle\alpha\alpha\rangle$, in contrast to (7). (Likewise the other density correlators also.) Therefore calculations of correlation functions at tree-level are more straightforward. In particular, Feynman diagrams reduce to the noise-vertex ones, such as (9), and do not include propagators such as (8). Secondly, it is now possible to use a response-field formalism to derive the corresponding real noises in the Langevin equations: the quadratic terms in the creation fields, $\mu(\tilde{\alpha} - \tilde{\pi})^2\alpha_0$ and $D(\nabla\tilde{\alpha})^2\alpha_0$, represent two noises respectively. Starting with the first of these, then since

$$e^{\int dt\, d^d x\, \mu\alpha_0(\tilde{\alpha}-\tilde{\pi})^2} \propto \int d\eta\, e^{\int dt\, d^d x\, (\tilde{\alpha}-\tilde{\pi})\eta - \eta^2/(4\mu\alpha_0)} \tag{24}$$

we recover a birth-and-death noise $\sqrt{2\mu a_0}\eta$ in the Langevin equation for $\rho_A$, and $-\sqrt{2\mu a_0}\eta$ in the Langevin equation for $\rho_P$, with $\eta$ unit white Gaussian noise. These two noises are equal and opposite, so there is no birth-and-death noise in the Langevin equation for the total density $\rho$. This must indeed be the case to conserve the total particle number. For the second term, notice that

$$e^{\int dt\, d^d x\, D\alpha_0(\nabla\tilde{\alpha})^2} \propto \int d\Xi\, e^{\int dt\, d^d x\, \Xi\cdot\nabla\tilde{\alpha} - \Xi^2/(4D\alpha_0)} \tag{25}$$

where $\Xi$ is unit white vectorial noise. Using integration by parts on the term $\Xi \cdot \nabla\tilde{\alpha}$, we find an additional conserved noise $\sqrt{2Da_0}\nabla \cdot \Xi$ in the Langevin equation for $\rho_A$.

Summarising the above, the Langevin equations at Gaussian level read

$$\frac{\partial \rho_A}{\partial t} = D\nabla^2\rho_A - \mu\rho_A + \kappa a_0(\rho - \rho_A - p_0) + \sqrt{2\mu a_0}\eta + \sqrt{2Da_0}\nabla \cdot \Xi \tag{26}$$

$$\frac{\partial \rho}{\partial t} = D\nabla^2\rho_A + \sqrt{2Da_0}\nabla \cdot \Xi \tag{27}$$

Here $\eta$ and $\Xi$ are unit white Gaussian noises as stated above. Directly solving for the equal-time structure factor from (26,27) confirms our result (19). Moreover one also finds that without the diffusive noise $\Xi$, the $a_0$ term in (19) is absent: hyperuniformity (now with $\varsigma = 2$) then also arises at $\xi < \infty$, that is, *away from criticality*.

Thus, neglecting diffusive noise alters the Gaussian-level predictions dramatically, although it is often neglected for C-DP and, by extension, RO on the basis that it is irrelevant in the RG sense [6, 15, 22]. Irrelevant variables that change correlation functions are known in the literature as *dangerously irrelevant* [23]. We return to this point in the RG setting for $d < 4$, in Section 5.2 below.

Meanwhile, our finding of an exponent $\varsigma = 2$ throughout the active phase matches results for particles with RO- or C-DP-like interactions in which the centre of mass of the density field is conserved [26]. This arises if particles move only by pairwise events in which the two particles are displaced by equal and opposite amounts [26]. Notably, without noise, (27) has an equivalent conservation law, as follows. The 'mass-moment' density field $\mathbf{p} = \rho \mathbf{r}$ obeys $\dot{p}_\alpha = -(\nabla_\beta J_\beta) r_\alpha = -\nabla_\beta (J_\beta r_\alpha) + J_\alpha$. For $\Xi = 0$ only, the current $\mathbf{J} = -D\nabla \rho_A$ in (27) is of pure gradient form, so that $\dot{p}_\alpha = -\nabla_\beta \Sigma_{\beta\alpha}$ with $\Sigma = \mathbf{J}\mathbf{r} + \mathbf{I}D\rho_A$ where $\mathbf{I}$ is the unit tensor. Hence the mass-moment $\int \mathbf{p} d\mathbf{r}$ in any region is conserved unless there are currents $\Sigma$ across its boundary [39]; for further discussions see [40]. In Section 2.2 we *defined* the RO universality class to exclude cases with this additional conservation law. It is an open question whether and how a Doi-Peliti theory can be constructed for the separate universality class describing the conserved centre-of-mass case, but as mentioned in Section 2.1, there is no simple way to delete the diffusive noise from our Doi-Peliti action for RO.

# 4 Perturbative Renormalisation Group Analysis

We now move onto the nonlinear theory that prevails in $d < d_c = 4$, working to one loop in perturbation theory or equivalently to first order in $\epsilon = 4 - d$. We mostly follow the standard procedure of using dimensional regularisation in $d = d_c - \epsilon = 4 - \epsilon$ dimensions [19, 41]. We start from an action constructed by a particular subset of allowed reactions, as employed at the Gaussian level in Section 3, that ensure the RO class is being studied. Doing so requires that there exists a critical FPM (degeneracy of fixed point action, assumption 1d in Section 2.2), and that perturbation theory is renormalisable (assumption 2a there).

## 4.1 The Nonlinear Action and its Scaling Fields

The full nonlinear bare Lagrangian, found after performing the shifts already detailed in Section 3, is

$$\mathbb{A} = -\tilde{a}(\partial_t - D\nabla^2)\breve{a} - \tilde{p}(\partial_t + \kappa a_0)\breve{p} + \kappa a_0 \tilde{a}\tilde{p} + \kappa a_0 p_0 (\tilde{a}^2 - \tilde{a}\tilde{p})$$
$$+ \kappa p_0 (\tilde{a}^2 - \tilde{a}\tilde{p})\breve{a} + \kappa a_0 (\tilde{a}^2 - \tilde{a}\tilde{p})\breve{p} + \kappa (\tilde{a} - \tilde{p})\breve{a}\breve{p} + \kappa (\tilde{a}^2 - \tilde{a}\tilde{p})\breve{a}\breve{p} \quad (28)$$

Here, the first line in (28) is the harmonic part of the action and the second line is the perturbative nonlinear part. The bare propagators can be read off from the action as

$$\begin{pmatrix} \langle \breve{a}(q,\omega)\tilde{a}(q',\omega')\rangle & \langle \breve{a}(q,\omega)\tilde{p}(q',\omega')\rangle \\ \langle \breve{p}(q,\omega)\tilde{a}(q',\omega')\rangle & \langle \breve{p}(q,\omega)\tilde{p}(q',\omega')\rangle \end{pmatrix} = \begin{pmatrix} -i\omega + Dq^2 & -\kappa a_0 \\ 0 & -i\omega + a_0\kappa \end{pmatrix}^{-1} \delta^d(q+q')\delta(\omega+\omega')$$

$$= \begin{pmatrix} \frac{1}{-i\omega + Dq^2} & \frac{\kappa a_0}{(-i\omega + Dq^2)(-i\omega + a_0\kappa)} \\ 0 & \frac{1}{-i\omega + a_0\kappa} \end{pmatrix} \delta^d(q+q')\delta(\omega+\omega')$$

$$(29)$$

At the RG fixed point, all relevant coupling constants will be renormalised and therefore can differ from their bare values shown above. Therefore we introduce names for these coupling constants and organize them systematically according to diagram topology. For example, the passive-to-active transmutation coupling will be denoted by $\tau_p$ and the passive mass by $\epsilon_p$; both have bare values of $\kappa a_0$. Similarly the active-to-passive transmutation coupling $\tau_a$ has zero bare value as does the active bare mass $\epsilon_a$. We show other couplings diagrammatically in (30) below, using the following (standard) rules for drawing Feynman diagrams in the Doi-Peliti formalism:

- Each annihilation field $\breve{a}$ (or $\breve{p}$) in the calculation of observable $\mathcal{O}$ is represented as a left end point of a leg.

- Each creation field $\tilde{a}$ (or $\tilde{p}$) in the calculation of observable $\mathcal{O}$ is represented as a right end point of a leg.

- Each interaction vertex in the action $\tilde{a}^k \breve{a}^l \tilde{p}^m \breve{p}^n$ is represented by a node with $l$ active propagators and $n$ passive propagators coming from the right, and $k$ active propagators and $m$ passive propagators going out from the left.

- Time flows from right to left, and hence the left-right order of lines must be obeyed in order to respect time ordering.

Drawing active fields as straight red lines and passive fields as blue wavy lines, we can now represent all the remaining interaction vertices present at bare level as amputated Feynman diagrams:

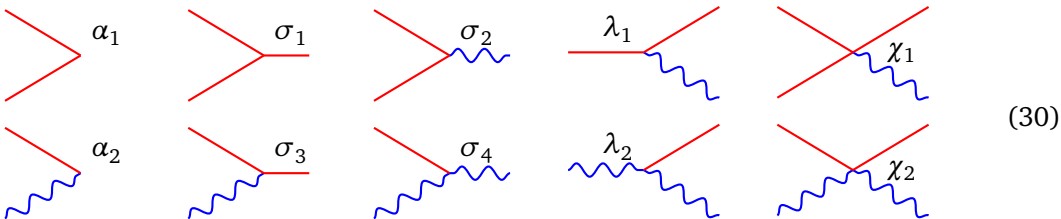

$$\tag{30}$$

Eq. (30) shows all the nonlinear coupling constants and vertices of the bare action. Other coupling constants could in general be generated at the RG fixed point without topological constraints on the corresponding vertices, that is, all vertices involving incoming and outgoing straight and wiggly lines are allowed in principle. However, we show in Sec. 4.3 that various other constraints on coupling constants are important in our theory for RO. Particularly important will be the following three interactions that are absent initially but could be generated at the RG fixed point:

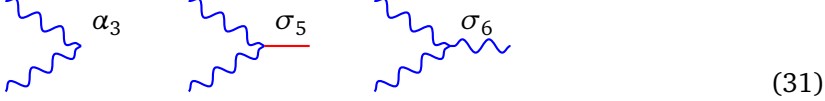

$$\tag{31}$$

These are the vertices that create non-renormalisable divergences as mentioned in Section 2 and Appendix B. Our assumption of renormalisability (2a in Section 2.2) indicates that these vertices must be suppressed beyond bare level via some type of cancellation at the critical RG fixed point.

In terms of a momentum scale $q \sim \zeta^1$, $x \sim \zeta^{-1}$ (as shorthand, we write $[q] = 1, [x] = -1$), one should usually be able to find scaling dimensions for all fields and coupling constants.

It should then be possible to find the engineering dimensions of the fields by power counting. However, rescaling of the fields is non-trivial in the RO model: creation and annihilation operators do not necessarily have the same engineering dimensions. This introduces two extra degrees of freedom for the scalings, that remain in question. For now, we assume that creation and annihilation fields have the same scalings at bare level, in which case $[\breve{a}(x,t)] = [\tilde{a}(x,t)] = [\breve{p}(x,t)] = [\tilde{p}(x,t)] = d/2$. The reason for this choice will become clearer in our discussions of anomalous dimensions in Section 5.1. The mass term in the passive propagator $-i\omega + \kappa a_0 := -i\omega + \epsilon_p$, determining the distance from criticality, then has dimension $[\epsilon_p] = 2$. The coupling constants also scale according to

$$[\tau_p] = [\alpha_i] = 2 \tag{32}$$

$$[\lambda_i] = [\sigma_i] = 2 - \frac{d}{2} \tag{33}$$

$$[\chi_i] = 2 - d \tag{34}$$

Therefore simply from looking at the scaling dimensions of the coupling constants, it can be concluded that the RO model has an upper critical dimension of $d_c = 4$, above which $\chi$, $\lambda$ and $\sigma$ vertices become irrelevant, and our results for the Gaussian approximation hold. Below 4 dimensions, $\lambda$ and $\sigma$ vertices become relevant. As noted already (and in Appendix B), if the interactions in (31) were present in the bare action, one would encounter loop corrections with both algebraic (not logarithmic) IR divergences (pointing to a higher $d_c = 6$ instead of 4), and non-renormalisable algebraic (not logarithmic) UV divergences.

## 4.2 Systematic Loop Counting

In $d = 4 - \epsilon$ dimensions, we perform perturbative renormalisation in 1-loop order. Due to the relatively large number of interaction vertices, a careful enumeration of all relevant 1-loop corrections is required. Here we use Euler's theorem for a connected planar graph

$$V - E + L = 1 \tag{35}$$

where $V$, $E$, $L$ represent the number of vertices, edges and loops respectively.

Consider loops constructed by the vertices $\alpha_i$, $\sigma_i$ and $\lambda_i$. We first require the number of additional outgoing legs to match the number of additional incoming legs. We note that an $\alpha$ interaction vertex introduces two extra outgoing legs; a $\sigma$ interaction vertex introduces two extra outgoing legs and one extra incoming leg; and a $\lambda$ interaction vertex introduces one extra outgoing leg and two extra incoming legs. From this we obtain (in an obvious notation)

$$2\#\alpha + \#\sigma = \#\lambda \tag{36}$$

After connecting up a selection of these interactions to form a 1-loop correction to some vertex of interest (with initial vertex number $V_0$ and edge number $E = E_0$) the final vertex number $V$ obeys

$$V = \#\alpha + \#\sigma + \#\lambda + V_0 \tag{37}$$

and the final edge number is

$$E = \#\alpha + \frac{3}{2}\#\sigma + \frac{3}{2}\#\lambda + E_0 \tag{38}$$

where the coefficients encode the fact that each new edge connects two vertices and that $\alpha$ vertices involve two edges but $\sigma$ and $\lambda$ involve three.

Substituting the expressions above into (35), we find that $\#\sigma + \#\lambda = 2$. Combined with equation (36), this says that each 1-loop correction must either be constructed by one $\sigma$ and one $\lambda$ interaction vertex, or two $\lambda$ and one $\alpha$ vertex. There are tens of these loops, but most of their corrections cancel eventually; we will show those that will contribute below. A full list that includes non-contributing diagrams is given in Appendix E.

### 4.2.1  $\lambda\sigma$-type Loops:

Topologically, all loops formed by one $\sigma$ and one $\lambda$ vertex look like the leftmost diagram below; from now on we use a black dashed line to represent either an active (red straight) or passive (blue wavy) propagator. The black dots represent vertices of $\sigma, \lambda$ or $\alpha$ type.

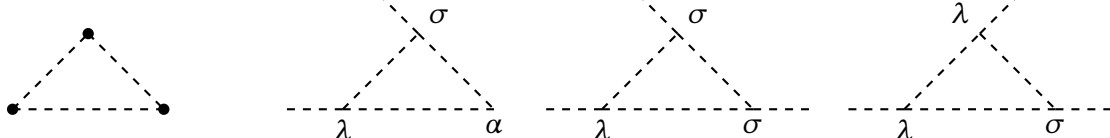

Each node is connected to external legs to form a correction to $\alpha$, $\sigma$ and $\lambda$ vertices, as shown respectively by the remaining three Feynman diagrams. We call these $\lambda\sigma$-type loops because the corresponding loop correction carries a prefactor of $\lambda\sigma$ in the $Z$-factors in later stages of the RG process. (See Section 4.4 for the standard definition of these $Z$-factors.) For example in the second diagram an $\alpha$ vertex contribution involves a $(\lambda, \sigma, \alpha)$ triangle, hence the $Z$-factor for the $\alpha$-vertex would contain the prefactor $\lambda\sigma\alpha/\alpha = \lambda\sigma$.

When substituting in the active/passive propagators for the black dashed lines, notice that some choices are algebraically (not logarithmically) divergent in $d = 4$ and hence individually non-renormalisable. Specifically, these are loop diagrams that include $\alpha_3$ and $\sigma_{5,6}$, as in (31), which are absent in the bare action but may generically be generated at the RG fixed point. Since we *assume* that the theory is renormalisable overall, as detailed in Section 2.2 (specifically assumptions 2a, 2c there), we implicitly assume that these vertices vanish at the fixed point. In consequence, the set of diagrams generating these vertices at the fixed point must cancel. Therefore all loop contributions at 1-loop order are those constructed without the couplings (31). These are shown below and named for future references:

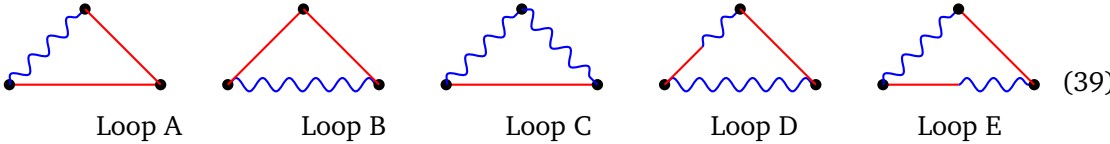

Loop A    Loop B    Loop C    Loop D    Loop E    (39)

Thereafter, the procedure for calculating the loop corrections contains two steps:

1. Calculate the loop integral for each of these five loops using dimensional regularization.

2. For each interaction vertex, identify the corresponding vertices used for the specific loops.

As an example, we consider Loop C shown in (39) as part of the first step. Using dimensional regularization, the loop integral is evaluated at vanishing external wavenumbers and frequencies,

$$
\,\hat{=}\, \int \frac{đ\omega đq}{(i\omega + \epsilon_p)(i\omega + \epsilon_p)(-i\omega + Dq^2 + \epsilon_a)}
$$

$$
= \int \frac{d^d q}{(2\pi)^d} \frac{1}{(Dq^2 + \epsilon_p + \epsilon_a)^2} = \frac{(\epsilon_p)^{d/2-2}}{D^{d/2}} \frac{\Gamma(2-\frac{d}{2})}{2^d \pi^{d/2}} \quad (40)
$$

Here, $\epsilon_{p,a}$ are the mass terms for passive and active propagator respectively, with bare values $\epsilon_p = \kappa a_0$ and $\epsilon_a = 0$. We will give their $Z$-factors, indicating how they are renormalised, in Section 4.4; in particular, we will show $Z_{\epsilon_a} = 1$, i.e. $\epsilon_a$ does not flow and stay at its bare

value zero (hence we drop it in the final expression above). In the second step, say we are considering the correction to the $\sigma_3$ vertex, or 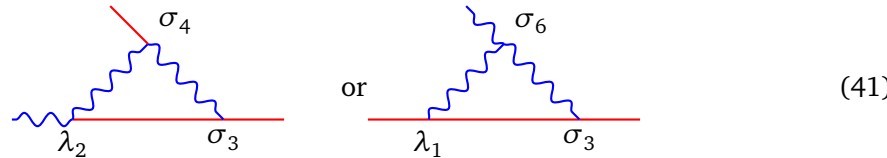 . There are, *a priori* two ways to attach this:

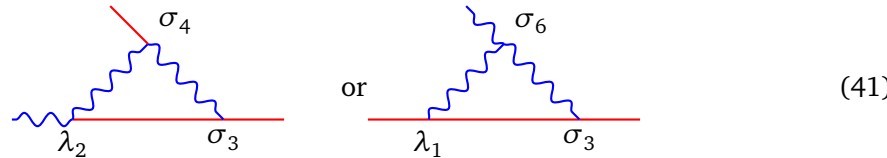

$$\tag{41}$$

However, the $\sigma_6$ vertex is eliminated by virtue of our renormalisability assumption (2a in Section 2.2). Therefore, in this example the contribution of the third loop in (39) to $\sigma_3$ has a prefactor of $\lambda_2 \sigma_3 \sigma_4$ with symmetry factor 1 coming from the first diagram above. The resulting contribution is $\lambda_2 \sigma_3 \sigma_4 \frac{(\epsilon_p)^{d/2-2}}{D^{d/2}} \frac{\Gamma(2-\frac{d}{2})}{2^d \pi^{d/2}}$. The exact same procedure is done for each of the five loops in (39), and for each interaction vertex $\alpha$, $\lambda$ and $\sigma$. In Section 4.4, we will show a full list of the resulting corrections after a further discussion of symmetries and effective coupling constants.

### 4.2.2 $\lambda\lambda\alpha$-type Loops:

We turn now to loops constructed by two $\lambda$ vertices and one $\alpha$ vertex. We first show topological diagrams for these loops and the attachment of external legs for correcting $\sigma$ and $\lambda$ vertices:

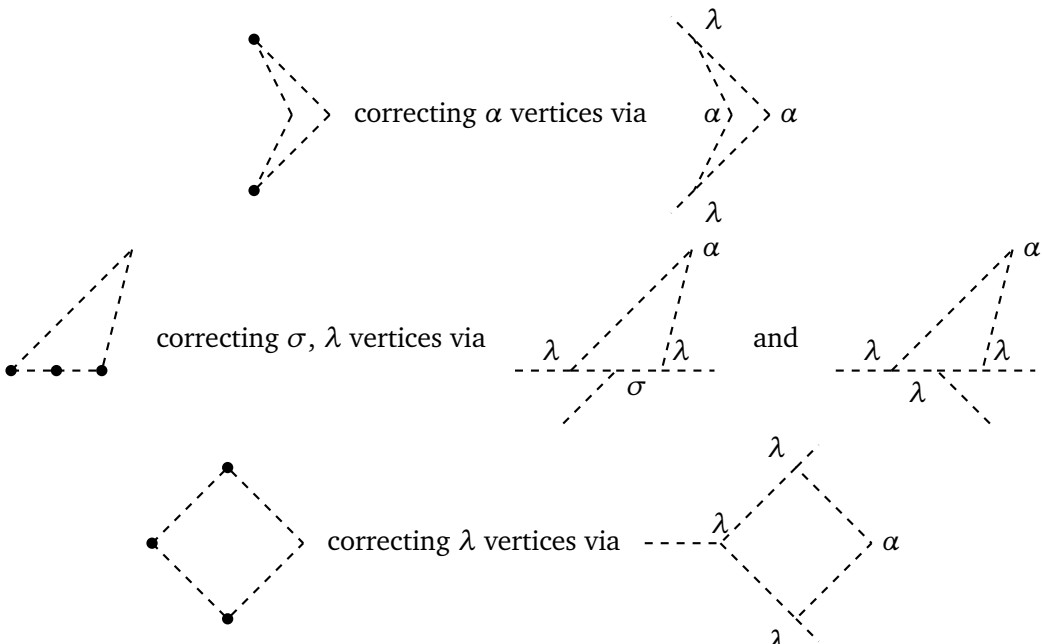

One can check that indeed each loop correction is topologically of the type $\lambda\lambda\alpha$. We find that there are only the three loops of 1-loop order that do not cancel with other loops (compared to the five in (39)) as follows:

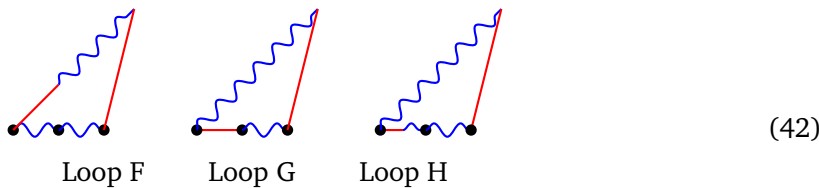

$$\tag{42}$$

Loop F     Loop G     Loop H

Again as an example, we apply dimensional regularization to Loop F:

$$\mathrel{\hat{=}} \tau_p \int \dd q \left[ \frac{1}{2Dq^2(Dq^2+\epsilon_p)^2(\epsilon_p-Dq^2)} \right.$$

$$\left. + \frac{1}{4(\epsilon_p)^2(Dq^2+\epsilon_p)(Dq^2-\epsilon_p)} \right] = \frac{1}{4}\frac{\tau_p}{\epsilon_p^2}\frac{(\epsilon_p)^{d/2-2}}{D^{d/2}}\frac{\Gamma(2-d/2)}{2^d\pi^{d/2}} \quad (43)$$

Then, following the second step described in 4.2.1, we consider how this loop contributes to (say) the $\sigma_4$ vertex . We find the exact same scenario as the $\lambda\sigma$-loop example elaborated above: with the prefactor $\lambda_2^2\alpha_2\sigma_4$, there is only one way to attach external propagators (hence symmetry factor 1) corresponding to the diagram below:

$$\mathrel{\hat{=}} \frac{1}{4}\frac{\lambda_2^2\alpha_2\sigma_4\tau_p}{\epsilon_p^2}\frac{(\epsilon_p)^{d/2-2}}{D^{d/2}}\frac{\Gamma(2-d/2)}{2^d\pi^{d/2}} \quad (44)$$

Note that there is no contribution from this type of loop in the corrections to either the $\alpha$ vertices (for which it is topologically impossible) or the $\sigma_1$, $\sigma_3$ vertices (where the leading order correction is 2-loop in general). Note also that in (44), the $\epsilon_p^2$ in the denominator does not induce higher order IR divergences despite going to zero close to the critical point. This is because in the numerator, both $\tau_p$ and $\alpha_2$ also have engineering dimensions $[\tau_p] = [\alpha_2] = [\epsilon_p] = 2$ (in contrast to being marginal). Therefore, at leading order, the prefactor in (44) remains nonsingular.

## 4.3 Identifying the Effective Coupling Constants

At bare level, there are ten nonlinear interaction vertices (see (30)) and ten interaction coupling constants, together with transmutation $\tau_p$, in the Doi-Peliti action (28). However, many of these arise from shifts in the creation and annihilation fields and they are further locked together because they originate from equal and opposite gain and loss terms of a master equation. In particular, all particle reactions modelled by the master equation conserve the total particle number. (There is no extinction, coagulation, $n$-tuple annihilation, or branching in the RO system.) The resulting relations (or 'symmetries') must be preserved under RG flow, thus reducing the number of independently relevant couplings. Moreover, any new couplings that are not in (28) but generated at the fixed point must obey these symmetries as well.

There are two ways to derive these field theoretic symmetries: in a bottom-up, non-perturbative, approach we start from the master equation and trace particle number conservation through the various transformations until we reach the coupling constants. In the alternative, perturbative approach, we consider the loop expansions of all couplings and identify which symmetries are maintained in the RG flow. We outline both approaches in the following, starting with the former.

Besides particle number conservation, there is another restriction in the RO class: there are infinitely many absorbing states. This implies that there cannot be a reaction – not even an effective one – in which all reactants are passive particles (the products can be any mix of passive and active particles as long as particle number is conserved).

### 4.3.1 Proof using Conservation Law:

In the second-quantized version of the master equation, the key observations are that (i) gain and loss terms contain equal numbers of annihilation operators and (ii) that the loss component will always contain an equal number of creation and annihilation operators while the gain will have equal numbers of them if and only if particle numbers are conserved. (For co-agulation, $n$-tuple annihilation or extinction processes, the gain term contains fewer creation than annihilation operators, whereas for branching processes, the gain term contains more creation than annihilation operators.) Furthermore, if a particle-number-conserving reaction contains a single change in particle type (single active to passive or vice versa), as is the case for our RO action, then loss and gain terms differ only in one creation operator while other creation operators are the same in both terms. This implies that the action must only contain terms that together have the factor $p^{\dagger} - a^{\dagger} = \widetilde{p} - \widetilde{a}$. This requirement remains true in the field theory where such operators are (essentially) replaced by their corresponding fields.

The factor $\widetilde{p} - \widetilde{a}$ is present in the bare action (28). In the RG flow, new couplings that emerge must still obey this symmetry: in combination with other such couplings they must maintain the factor $\widetilde{p} - \widetilde{a}$ across all the interaction terms. The couplings that group together in this way attach to vertices that have identical annihilation field legs (in both number and types) and also the same number, but not necessarily same type, of Doi-shifted creation field legs. This requirement implies that

$$\lambda_1 = -\lambda_2 \tag{45a}$$
$$\epsilon_p = \tau_p \tag{45b}$$
$$\epsilon_a = \tau_a(= 0) \tag{45c}$$
$$\alpha_1 = -(\alpha_2 + \alpha_3) \tag{45d}$$
$$\sigma_1 = -(\sigma_3 + \sigma_5) \tag{45e}$$
$$\sigma_2 = -(\sigma_4 + \sigma_6) \tag{45f}$$
$$\chi_1 = -(\chi_2 + \chi_3) \tag{45g}$$

### 4.3.2 Diagrammatical Proof by Comparing Loop Corrections:

The above line of argument uses a microscopic conservation law on total particle number to constrain the nonlinearities to all orders. However, the results (45) can alternatively be found entirely at field-theoretic level by analyzing the loop expansions of the couplings.

We start with the more straightforward case: establishing the first two symmetries in (45). Here, there is only one outgoing external leg, allowing easy comparison of the loop corrections. There is a *one-to-one correspondence* of loops, with prefactor ratios that cannot change. This means that once we calculate the $Z$-factors (defined by dividing the loop correction by the corresponding vertex, see Section 4.4), these are exactly the same for each of the first two pairs in (45). A diagrammatic illustration is shown below for $\lambda\sigma$-type loops correcting $\lambda$ vertices.

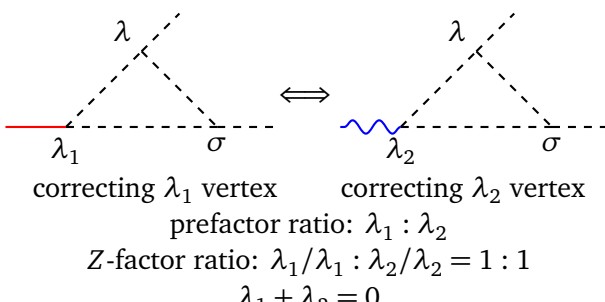

correcting $\lambda_1$ vertex    correcting $\lambda_2$ vertex
prefactor ratio: $\lambda_1 : \lambda_2$
$Z$-factor ratio: $\lambda_1/\lambda_1 : \lambda_2/\lambda_2 = 1 : 1$
$\lambda_1 + \lambda_2 = 0$

Establishing the other symmetries in (45) is slightly less straightforward. Starting with $\lambda\sigma$-type loops, we find that for Loops A, C, D, E (39), the corresponding $Z$-factor contribution is identical. Take Loop C as an example,

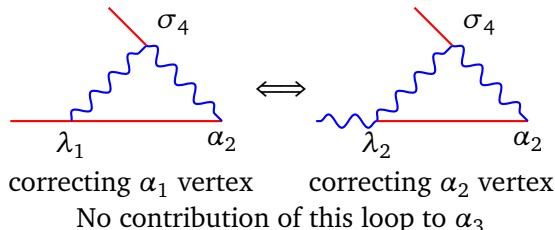

correcting $\alpha_1$ vertex      correcting $\alpha_2$ vertex
No contribution of this loop to $\alpha_3$

Since $\lambda_1 = -\lambda_2$, and the symmetry factor for both Feynman diagrams is one, the prefactors for the corrections from this loop are equal and opposite. Since at bare level, $\alpha_1 + \alpha_2 = \alpha_3 = 0$ and $\alpha_3$ is not corrected by this loop, the symmetry $\alpha_1 + \alpha_2 + \alpha_3 = 0$ is preserved.

Subtlety arises however for loops that do also correct $\alpha_3$ at 1-loop order. For $\lambda\sigma$-type loops, this only happens for Loop B in (39). Diagrammatically,

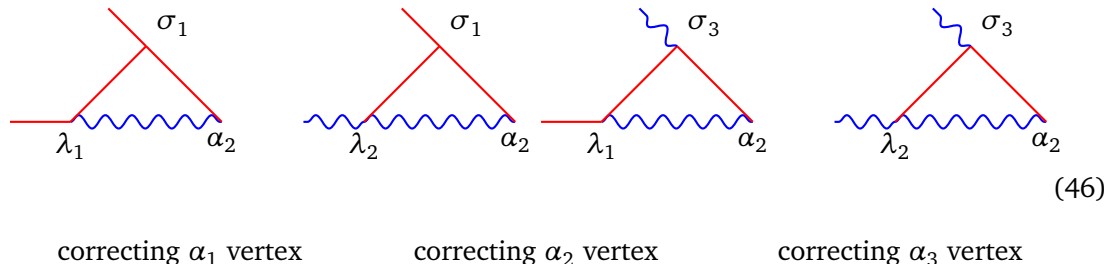

$$(46)$$

correcting $\alpha_1$ vertex        correcting $\alpha_2$ vertex        correcting $\alpha_3$ vertex

Here the first two Feynman diagrams have a symmetry factor of two (due to the use of $\sigma_1$ vertex: the two outgoing legs could interchange), and the other two have a symmetry factor of one. Hence if we consider the correction of this loop to the quantity $(\alpha_1 + \alpha_2 + \alpha_3)$, the contribution from this loop is $(2\lambda_1\sigma_1 + 2\lambda_2\sigma_1 + \lambda_1\sigma_3 + \lambda_2\sigma_3)\alpha_2 \times [\text{loop integral}]$. Therefore, the independently established symmetry $\lambda_1 = -\lambda_2$ ensures that again there are no loop corrections to $\alpha_1 + \alpha_2 + \alpha_3$. It follows that even when a loop corrects also the absent vertices in the action, this does not change the fact that $\alpha_1 + \alpha_2 + \alpha_3 = 0$. A similar check can be done on all three $\lambda\lambda\alpha$-type loops, completing our verification, at 1-loop order, of all the symmetries listed in (45).

### 4.3.3 Effective Coupling Constants:

As discussed earlier in assumptions 2a, 2c, 2d, vertices (31) cannot be generated at the fixed point, thanks to a cancellation mechanism which we *assume* to be broadly similar to the tri-critical Ising model. Nevertheless, we do observe loop corrections to $\alpha_3$ as we show in (46). These corrections cannot be absorbed into the $Z$-factors associated with $\alpha_3$, as $\alpha_3$ is absent at the RG fixed point; however, ignoring such contributions completely would undermine the basic tenets of the RG, which requires each UV divergence to be taken into account by $Z$-factors, and violate the symmetry expressed in (46). Thus there has to be a systematic way of absorbing these loop corrections into other $Z$-factors, ensuring that the contributions to the bare correlators are properly accounted for.

To elaborate this point, we examine how loop corrections to *vertices* contribute to *correlators*. Normally, loop corrections at vertex level are encoded in the corresponding $Z$-factors, diagrammatically representing the loops with patterned circles:

$$Z_{\alpha_1} = 1 + \text{⬤}, \quad Z_{\alpha_2} = 1 + \text{⬤}, \quad Z_{\alpha_3} = 1 + \text{⬤} \tag{47}$$

Correlators are calculated by appending external propagators to the vertices. At tree-level, the active-active correlator $\langle \breve{a}\breve{a} \rangle \; \hat{=} \; \text{⟩}\, \alpha_1 \; + \; \text{⟩}\, \alpha_2 \; + \; \text{⟩}\, \alpha_3$ , active-passive correlator $\langle \breve{a}\breve{p} \rangle \; \hat{=} \; \text{⟩}\, \alpha_2 \; + \; \text{⟩}\, \alpha_3$ , and passive-passive correlator $\langle \breve{p}\breve{p} \rangle \; \hat{=} \; \text{⟩}\, \alpha_3$ . Note that transmutation only exists in one direction ($\text{—}\!\sim\!\text{—}$ is present but not $\sim\!\text{—}$).

The tree-level contributions correspond to the '1' in the $Z$-factors, while loop corrections modify this factor. For example, the active-active correlator is composed of three contributions, each corresponding to $Z_{\alpha_1}$, $Z_{\alpha_2}$, and $Z_{\alpha_3}$. Diagrammatically,

$$\langle \breve{a}\breve{a} \rangle \; \hat{=} \; \text{⟩} + \text{⟩} + \text{⟩} + \text{⬤} + \text{⬤} + \text{⬤} \tag{48a}$$

$$\langle \breve{a}\breve{p} \rangle \; \hat{=} \; \text{⟩} + \text{⟩} + \text{⬤} + \text{⬤} \tag{48b}$$

$$\langle \breve{p}\breve{p} \rangle \; \hat{=} \; \text{⟩} + \text{⬤} \tag{48c}$$

In general, there should be a similarly straightforward correspondence between $Z$-factors for vertices and corrections to correlators. However, as discussed above, our assumption of renormalisability dictates that $\alpha_3$, and hence $Z_{\alpha_3}$, cannot appear at the fixed point. We thus have two options for the $Z_{\alpha_3}$: either absorb the correction in $Z_{\alpha_1}$ or $Z_{\alpha_2}$. In the latter case,

$$Z_{\alpha_1} = 1 + \text{⬤}, \quad Z_{\alpha_2} = 1 + \text{⬤} + \text{⬤} \tag{49}$$

and the correlators obey

$$\langle \breve{a}\breve{a} \rangle \; \hat{=} \; \text{⟩} + \text{⟩} + \text{⬤} + \text{⬤} + \text{⬤} \tag{50a}$$

$$\langle \breve{a}\breve{p} \rangle \; \hat{=} \; \text{⟩} + \text{⬤} + \text{⬤} \tag{50b}$$

capturing all loop corrections in (48). In the former case, we instead assign the correction in $Z_{\alpha_3}$ to $Z_{\alpha_1}$, diagrammatically

$$Z_{\alpha_1} = 1 + \text{⬤} + \text{⬤}, \quad Z_{\alpha_2} = 1 + \text{⬤} \tag{51}$$

However, in the active-passive correlator, the loop correction $\text{⬤}$ is no longer accessible since the vertex $\alpha_1$ does not enter the correlator associated with $\langle \breve{a}\breve{p} \rangle$. We thus conclude that the systematic way of treating $Z_{\alpha_3}$ is to absorb it into the $Z$-factor for $\alpha_2$. We note that this unique choice is a direct result of transmutation working in only one direction.

## 4.4 Beta Functions and Critical Exponents

We now define the various $Z$-factors (some of which were already discussed) indicating how fields and interaction vertices should be renormalised:

$$D_R = Z_D D \quad \breve{a}_R = Z_{\breve{a}}^{1/2} \breve{a} \quad \tilde{a}_R = Z_{\tilde{a}}^{1/2} \tilde{a} \quad \breve{p}_R = Z_{\breve{p}}^{1/2} \breve{p} \quad \tilde{p}_R = Z_{\tilde{p}}^{1/2} \tilde{p} \tag{52}$$

$$\alpha_R = Z_\alpha \alpha \zeta^{-2} \quad \lambda_R = A_d^{1/2} Z_\lambda \lambda \zeta^{(d-4)/2} \quad \sigma_R = A_d^{1/2} Z_\sigma \sigma \zeta^{(d-4)/2} \tag{53}$$

$$\text{with } A_d := \frac{\Gamma(3 - d/2)}{2^{d-1}\pi^{d/2}} \quad \text{(a constant arising from angular integrals)} \tag{54}$$

Here, $\Gamma$ represents the Gamma function. (Elsewhere in this paper where $\Gamma$ denotes a vertex functions instead, it is written with subscripts indicating the corresponding vertex or propagator.) Addressing first the field renormalisations, observe that the active propagator is corrected by the following loop diagrams:

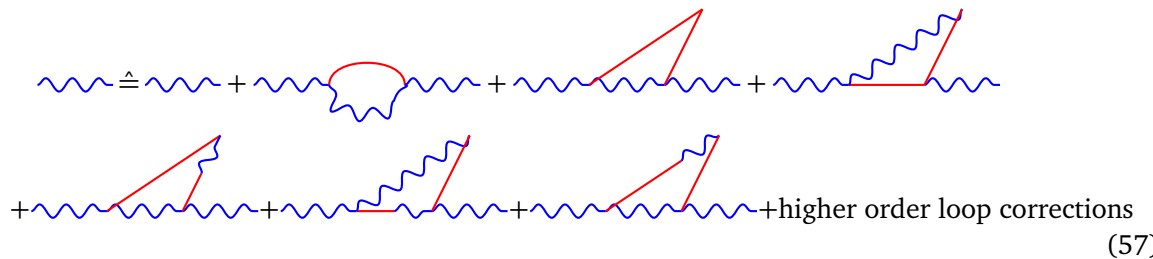

$$\tag{55}$$

This corresponds to the corrections to vertex functions

$$\Gamma_{\text{active propagator}}(q, \omega) = i\omega + Dq^2 + \epsilon_a + \lambda_1 \sigma_3 \int_k \frac{1}{Dk^2 + \epsilon_a + \epsilon_p + i\omega}$$
$$+ \lambda_1 \lambda_2 \alpha_2 \int_k \frac{1}{(D(q-k)^2 + \epsilon_a + \epsilon_p)(D(q-k)^2 + \epsilon_a + \epsilon_p + i\omega)} \tag{56}$$

This vanishes at vanishing wavenumber $q$ and frequency $\omega$. Also, only here we have included $\epsilon_a$ as we will use (56) to prove that $\epsilon_a$ retain its bare value zero throughout the flow (it is hence dropped in all other loop integrals). Similarly, the passive propagator is corrected by:

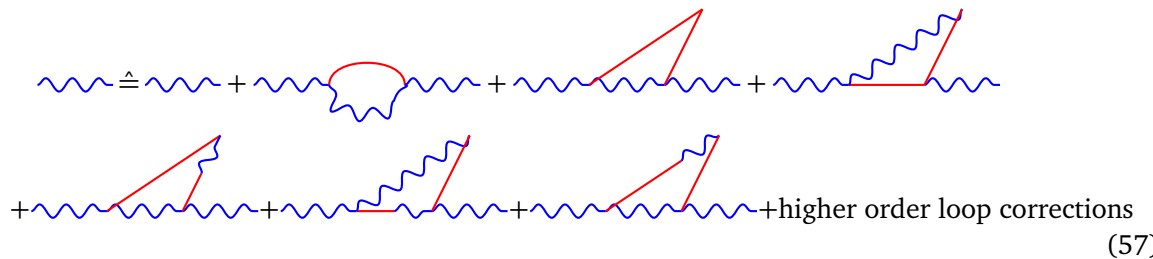

$$\tag{57}$$

corresponding to

$$\Gamma_{\text{passive propagator}}(q, \omega) = i\omega + \epsilon_p + \lambda_2 \sigma_4 \int_k \frac{1}{Dk^2 + \epsilon_p + i\omega} + 2\lambda_2^2 \alpha_1 \int_k \frac{1}{(Dk^2 + \epsilon_p + i\omega)(2Dk^2)}$$
$$+ \lambda_1 \lambda_2 \alpha_2 \int_k \frac{1}{(Dk^2 + \epsilon_p)(i\omega + D(q+k)^2 + \epsilon_p)} + \lambda_2^2 \tau_p \alpha_2 \int_k \frac{1}{2Dk^2(Dk^2 + \epsilon_p)(Dk^2 + \epsilon_p + i\omega)}$$
$$+ \lambda_2^2 \tau_p \alpha_2 \int_k \frac{1}{(Dk^2 + \epsilon_p)(i\omega + 2\epsilon_p)(D(q+k)^2 + \epsilon_p + i\omega)}$$
$$+ \lambda_2^2 \tau_p \alpha_2 \int_k \frac{1}{(\epsilon_p - Dk^2)(2Dk^2)(Dk^2 + \epsilon_p + i\omega)} + \frac{1}{(Dk^2 - \epsilon_p)(Dk^2 + \epsilon_p)(i\omega + 2\epsilon_p)} \tag{58}$$

The $Z$-factors for the fields are evaluated at normalization point $q = \omega = 0$, $\epsilon_p = \zeta^2$ to lowest order in nonlinearities. We use

$$\int \frac{d^d k}{(2\pi)^d} \frac{1}{(\epsilon_p + k^2)^s} = \frac{\Gamma(s - d/2)}{2^d \pi^{d/2} \Gamma(s)} \epsilon_p^{-s + d/2} \tag{59}$$

and apply this to the derivatives $\partial \Gamma_{\text{propagators}} / \partial(i\omega)$. For the active propagator field renormalisation,

$$
\begin{aligned}
Z_{\check{a}}^{1/2} Z_{\tilde{a}}^{1/2} &= \frac{\partial \Gamma_{\text{active propagator}}(q, \omega)}{\partial i\omega} \bigg|_{(q,\omega)=(0,0)} \\
&= 1 - \lambda_1 \sigma_3 \int_k \frac{1}{(Dk^2 + \epsilon_p)^2} - \lambda_1 \lambda_2 \alpha_2 \int_k \frac{1}{(Dk^2 + \epsilon_p)^3} \\
&= 1 - \frac{\lambda_1 \sigma_3}{D^{d/2}} \frac{A_d \zeta^{-\epsilon}}{\epsilon} - \frac{\lambda_1 \lambda_2 \alpha_2}{\epsilon_p} \cdot \text{finite integral} = 1 - \frac{\lambda_1 \sigma_3}{D^{d/2}} \frac{A_d \zeta^{-\epsilon}}{\epsilon}
\end{aligned}
\tag{60}
$$

Note the last term is UV-finite and hence dropped from the $Z$-factor. This is known as minimal subtraction. It also does *not* induce higher order IR divergences being absorbed by the $\alpha$-term in the numerator (as was shown in (44)). A similar calculation can be done for the passive propagator to obtain

$$Z_{\check{p}}^{1/2} Z_{\tilde{p}}^{1/2} = \frac{\partial \Gamma_{\text{passive propagator}}(q, \omega)}{\partial i\omega} = 1 - \frac{\lambda_2 \sigma_4}{D^{d/2}} \frac{A_d \zeta^{-\epsilon}}{\epsilon} - \frac{1}{2} \frac{\lambda_2^2 \alpha_2 \tau_p}{\epsilon_p^2} \frac{1}{D^{d/2}} \frac{A_d \zeta^{-\epsilon}}{\epsilon}. \tag{61}$$

Then applying similar calculations to the following derivatives gives

$$Z_{\check{a}}^{1/2} Z_{\tilde{a}}^{1/2} Z_D = \frac{1}{D} \frac{\partial \Gamma_{\text{active propagator}}(q, \omega)}{\partial q^2} \bigg|_{(q,\omega)=(0,0)} = 1 \tag{62}$$

$$Z_{\check{a}}^{1/2} Z_{\tilde{a}}^{1/2} Z_{\epsilon_a} = \frac{\partial \Gamma_{\text{active propagator}}(q, \omega)}{\partial \epsilon_a} \bigg|_{(q,\omega)=(0,0)} = 1 - \frac{\lambda_1 \sigma_3}{D^{d/2}} \frac{A_d \zeta^{-\epsilon}}{\epsilon} \tag{63}$$

$$Z_{\check{p}}^{1/2} Z_{\tilde{p}}^{1/2} Z_{\epsilon_p} = \frac{\partial \Gamma_{\text{passive propagator}}(q, \omega)}{\partial \epsilon_p} \bigg|_{(q,\omega)=(0,0)} = 1 - \frac{\lambda_2 \sigma_4}{D^{d/2}} \frac{A_d \zeta^{-\epsilon}}{\epsilon} - \frac{\lambda_2^2 \alpha_2 \tau_p}{\epsilon_p^2} \frac{1}{D^{d/2}} \frac{A_d \zeta^{-\epsilon}}{\epsilon} \tag{64}$$

Hence, we have found the $Z$-factors for the mass in the active propagator $\epsilon_a$, diffusion constant $D$ and the distance to the critical point $\epsilon_p$,

$$Z_{\epsilon_a} = 1, \quad Z_D = 1 + \frac{\lambda_1 \sigma_3}{D^{d/2}} \frac{A_d \zeta^{-\epsilon}}{\epsilon}, \quad Z_{\epsilon_p} = 1 - \frac{\lambda_2^2 \alpha_2 \tau_p}{2 \epsilon_p^2 D^{d/2}} \frac{A_d \zeta^{-\epsilon}}{\epsilon} \tag{65}$$

Note that the $Z$-factor for the mass in active propagator to be identically 1 implies that it is not renormalised; this confirms we can safely write the active propagator in its bare, massless form as done in several places above. The scalings of $D$ and $\epsilon_p$ give rise to the dynamic exponent $z$ and correlation length exponent $\nu_\perp$ respectively.

As observed from the $Z$-factors, there are three renormalised dimensionless effective cou-

plings, namely

$$u_R = \frac{\lambda_1 \sigma_3}{D^{d/2}} A_d \zeta^{-\epsilon} \frac{Z_{\lambda_1} Z_{\sigma_3}}{Z_D^{d/2}} \tag{66}$$

$$v_R = \frac{\lambda_2 \sigma_4}{D^{d/2}} A_d \zeta^{-\epsilon} \frac{Z_{\lambda_2} Z_{\sigma_4}}{Z_D^{d/2}} \tag{67}$$

$$w_R = \frac{\lambda_2^2 \alpha_2 \tau_p}{\epsilon_p^2 D^{d/2}} A_d \zeta^{-\epsilon} \frac{Z_{\lambda_2}^2 Z_{\alpha_2}}{Z_{\epsilon_p} Z_D^{d/2}} \tag{68}$$

In the Gaussian theory where all nonlinearities $\lambda$'s and $\sigma$'s are suppressed, the renormalised effective couplings $u_R, v_R, w_R$ vanish. Below the upper critical dimension $d_c = 4$, they acquire non-zero values. Importantly, we have defined *combinations* of nonlinearities as effective couplings, because it is only these combinations, not individual nonlinearities, that construct loops and enter beta functions. Such construction of a renormalised effective coupling appears in directed percolation as well [19]. For the $\lambda\sigma$-type loops, there is only one degree of freedom for $\lambda$ (since $\lambda_1 + \lambda_2 = 0$), and two degrees of freedom for $\sigma$ (since $\sigma_1 + \sigma_3 = 0, \sigma_2 + \sigma_4 = 0$). Hence there are two effective couplings of this loop type, namely $u_R$ and $v_R$. For $\lambda\lambda\alpha$-type loops, $\alpha$ only has one degree of freedom ($\alpha_1 + \alpha_2 = 0$), implying that there can only be one effective coupling of this loop type, namely $w_R$.

RG fixed points are found by setting the beta functions (namely, flow functions for *effective coupling constants*) for the couplings in (66)-(68) equal to zero. As promised in Section 4.2, we present in Appendix D the full list of one-loop corrections to vertices. As shown there, the final beta functions for the effective coupling constants read

$$\beta_u = \zeta \frac{\partial}{\partial \zeta}\bigg|_0 u_R = u_R(\gamma_{\lambda_1} + \gamma_{\sigma_3} - \frac{d}{2}\gamma_D) = u_R\left[-\epsilon - 3u_R - 3v_R - \frac{1}{2}w_R + \mathcal{O}(\epsilon^2)\right] \tag{69}$$

$$\beta_v = \zeta \frac{\partial}{\partial \zeta}\bigg|_0 v_R = v_R(\gamma_{\lambda_2} + \gamma_{\sigma_4} - \frac{d}{2}\gamma_D) = v_R\left[-\epsilon - 2u_R - 4v_R - \frac{1}{2}w_R + \mathcal{O}(\epsilon^2)\right] \tag{70}$$

$$\beta_w = \zeta \frac{\partial}{\partial \zeta}\bigg|_0 w_R = w_R(2\gamma_{\lambda_2} + \gamma_{\alpha_2} - \gamma_{\epsilon_p} - \frac{d}{2}\gamma_D) = w_R\left[-\epsilon - 4u_R - 5v_R - \frac{3}{2}w_R + \mathcal{O}(\epsilon^2)\right] \tag{71}$$

Notice that there is no need to settle degrees of freedom in field renormalisation at this point, since they always show up in pairs of $(\tilde{a}, \breve{a})$ and $(\tilde{p}, \breve{p})$ in the beta functions of the effective coupling constants. Hence the non-Gaussian RG fixed point for the RO universality class obeys the following linear system of equations

$$3u_R + 3v_R + \frac{1}{2}w_R = -\epsilon \tag{72a}$$

$$2u_R + 4v_R + \frac{1}{2}w_R = -\epsilon \tag{72b}$$

$$4u_R + 5v_R + \frac{3}{2}w_R = -\epsilon \tag{72c}$$

Solving these gives $u_R^* = v_R^* = -2\epsilon/9$ and $w_R^* = 2\epsilon/3$. Substituting these results in (65) produces the scaling dimensions for the diffusion constant $[D] = u_R^* = -\frac{2}{9}\epsilon$ and order parameter $[\epsilon_p] = 2 - 1/2w_R^* = 2 - \frac{1}{3}\epsilon$, and thus the dynamic critical exponent $z$ and correlation length exponent $1/\nu_\perp$:

$$D \sim \epsilon_p^{\nu_\perp(z-2)} \Rightarrow z = 2 - \frac{2}{9}\epsilon \tag{73a}$$

$$\xi \sim \epsilon_p^{-\nu_\perp} \Rightarrow \frac{1}{\nu_\perp} = 2 - \frac{1}{3}\epsilon \tag{73b}$$

These match to order $\epsilon$ the results found via the mapping onto the quenched Edwards-Wilkinson model (q-EW) via functional RG [14, 16], in line with our assumption 1b in Section 2.2.

A linear stability analysis reveals that the RO fixed point has two stable directions and one unstable direction in the space of effective couplings $(u_R, v_R, w_R)$. Crucially, one of the attractive directions aligns with the vector $(u_R^*, v_R^*, w_R^*)$, indicating the existence of a RG trajectory of the running couplings that connects the Gaussian fixed point in the UV limit to the RO fixed point in the IR limit which, to $O(\epsilon)$, is a straight line in coupling space. This ensures that the RO fixed point, though not fully attractive, can still be accessed perturbatively by appropriately fine-tuning parameters (or, in effect, enforcing cancellation among the algebraic divergences referred to previously). This again bears a high resemblance to the tricritical Ising fixed point, which is more unstable than the usual Wilson-Fisher fixed point which belongs to a universality class of higher upper critical dimension [42, 43].

There are two further things to note. Firstly, these results give us the sum of anomalous dimensions of the annihilation and creation of fields $[\tilde{a}(x,t)\breve{a}(x,t)] = d + 2\epsilon/9$, $[\tilde{p}(x,t)\breve{p}(x,t)] = d - \epsilon/9$. They do not give the dimensions of $\tilde{a}, \breve{a}, \tilde{p}, \breve{p}$ separately; these will be discussed in the next Section. Secondly, the shifts of the annihilation fields ($a_0$ and $p_0$) cannot be expressed as any combination of the effective coupling constants $u_R, v_R$ and $w_R$. Since all universal behaviour to order $\mathcal{O}(\epsilon)$ should be defined by these RG fixed point values, we deduce that these shifts are nonuniversal. This is not surprising: as previously explained, they describe initialisation of the system in the far past.

# 5 Hyperuniformity and its Exponent

In this Section, we first use our *assumption* that hyperuniformity emerges at the RO critical point (as opposed to divergent fluctuations: see assumption (3) in Section 2.2 above) to *derive* the hyperuniformity exponent $\varsigma$ to order $\epsilon$. We highlight how, in striking similarity to the Gaussian theory (Section 3.2), hyperuniformity emerges by cancellation of separately diverging fluctuations for the active and passive particle species. Thereafter we discuss the implications and physical interpretation of our result for the exponent $\varsigma$.

## 5.1 The Hyperuniformity Exponent $\varsigma$

To find the remaining exponents $\beta$ and $\varsigma$ we need the dimensions of each field variable separately. Below we show that only one choice (the same as would arise from assuming rapidity reversal in passive particles near criticality [44]) is consistent with hyperuniformity, with all other choices giving *divergent* rather than zero low-$q$ fluctuations at criticality. It is, of course, not unusual for physical knowledge concerning a critical point to resolve ambiguities in an RG calculation, but intriguing that here the requirement of hyperuniformity itself is sufficient to do so.

To see how this works, let us consider $S(q)$. The structure factor for the total density consists of three parts: the active covariance (7), the passive covariance (12) and the active-passive covariance (14). While the complete form for the structure factor is lengthy, we can make the following observations:

- Terms that involve transmutations, such as $\langle \breve{a}\tilde{p}' \rangle$, vanish in the equal-time limit. (The notation is such that a prime indicates that the field has argument $(-q, t)$, whereas without the prime the argument is $(q, t)$. This is similar to the notation in Section 3.2 but the temporal arguments are taken to be equal.) Physically this is because the insertion of an active/passive density cannot instantaneously affect the density of the other species.

- Observables that involve three field operators such as $\langle \breve{a}\tilde{a}'\breve{a}'\rangle$ also vanish. This is because diagrams that contribute to these, such as ———×, though allowed in Doi-Peliti field theory, give contributions that contain prefactors of negative powers of $\xi$ hence giving zero contributions.

- Terms of the form of $\langle \breve{a}\rangle\langle \breve{a}\rangle$ cancel collectively due to conservation of total density across the three correlators: $\langle \breve{a}\rangle\langle \breve{a}\rangle + 2\langle \breve{a}\rangle\langle \breve{p}\rangle + \langle \breve{p}\rangle\langle \breve{p}\rangle = 0$ since $\langle \breve{a}\rangle + \langle \breve{p}\rangle = 0$.

Using these observations, the contributing parts can be gathered to give the structure factor for the total density $\rho = \rho_A + \rho_P$ as:

$$\delta^d(0)S(q) := \langle\rho(q,t)\rho(-q,t)\rangle = a_0\left\langle \breve{a}\tilde{a}'\right\rangle + \left\langle \breve{a}\breve{a}'\right\rangle + \left\langle \breve{a}\breve{p}'\right\rangle + \left\langle \breve{p}\breve{a}'\right\rangle + \left\langle \breve{p}\breve{p}'\right\rangle + p_0\left\langle \breve{p}\tilde{p}'\right\rangle \quad (74)$$

Here all the correlators are equal-time, with $q, -q$ arguments suppressed for clarity. Evaluating all right hand side correlators at momenta $q$ and $q'$ produces a factor of $\delta^d(q+q')$, which gives rise to $\delta^d(0)$ in Eq. (74). The first two terms represent the active-active density correlator, the next two the active-passive cross correlations, and the final term the passive-passive density correlator. Our RG approach gives no information on amplitude ratios for these terms, so that any cancellations among them cannot be found by comparing prefactors (in contrast to the Gaussian case in (19)). However, their $q$ dependences at criticality are directly set by the scaling dimensions of the four fields, $\breve{a}, \tilde{a}, \breve{p}, \tilde{p}$, and this will be enough for us.

From the anomalous dimensions reported after (72c) above for $[\breve{a}\tilde{a}], [\breve{p}\tilde{p}]$, we observe that at criticality ($\xi \to \infty$) the first of the six terms in (74) scales as $S_1 \sim a_0 q^{2\epsilon/9}$ and the last as $S_6 \sim p_0 q^{-\epsilon/9}$ (after factoring out the $\delta^d(0)$ divergence). The conservation of total particle density (and thus the symmetries proved in 4.3) implies that active and passive fluctuations must scale alike, *i.e.*, the anomalous dimensions for active and passive annihilation fields must match: $[\breve{a}] = [\breve{p}]$. Hence the terms $S_{2,3,4,5}$ all share the same scaling behaviour. Note that, as mentioned already in Section 3.2, the shift $a_0$ no longer vanishes at criticality as it does in the Gaussian limit. Moreover (see Section 4.4) $a_0$ cannot be written as a combination of the $u, v, w$ effective coupling constants, nor of the particle field operators, arising in the action (2) or its shifted counterpart. This means that $a_0$ cannot acquire an anomalous dimension at the fixed point: it merely acts as a non-universal amplitude. (The same is true of $p_0$, as can separately be confirmed by requiring the exponent $\beta$ to match the q-EW result.) Accordingly, for the system to be hyperuniform rather than divergently fluctuating at low $q$ (assumption 3 in Section 2.2), $S_6$ must be cancelled by some combination of the terms $S_{2,3,4,5}$. This requirement *alone* fixes the anomalous dimensions of the fields as $\eta_{\breve{a}} = \eta_{\breve{p}} = \eta_{\tilde{p}} = -\epsilon/18$ and $\eta_{\tilde{a}} = 5\epsilon/18$: only then can all negative powers of $q$ cancel in (74).

Although the terms involved are now divergent rather than finite at $q \to 0$, this cancellation resembles the one found (albeit via a different order of limits) for the Gaussian model in (19) and Fig. 1. As found there, strongly fluctuating active and passive quantities must cross-correlate such that their sum is hyperuniform. Moreover, since every term $S_{i>1}$ in (74) involve just two fields, each has a pure scaling behavior $\xi^0 q^{-\epsilon/9} F_i(q\xi)$ with $F_i(s)$ regular at large $s$ [19]. Hyperuniformity then requires $\sum_{i=2}^{6} F_i(\infty) = 0$, and whatever remains after this cancellation vanishes at criticality where $\xi \to \infty$. (This reasoning would not hold if the individual $S_i$ were, like $S$, correlators of sums of fields [45].) Accordingly, given that $a_0$ is not singular as detailed above, the hyperuniformity exponent governing $S(q)$ at the critical point can be read off from $S_1$ as $\varsigma = 2\epsilon/9$.

As we have emphasised, the critical regime exhibits hyperuniformity of the total density $\rho$, but not of the active and passive densities separately. Instead, the correlators for these each diverge as $q^{-\epsilon/9}$, confirming a previously known value of $2 - \epsilon/9$ [14] for the exponent $\eta_\perp$ defined via $S_{AA} \sim = q^{-2+\eta_\perp}$ [2]. The anomalous dimensions determined above also imply $\beta = \nu_\perp(d/2 + \eta_{\breve{a}}) = 1 - \epsilon/9$, again matching the results found via the mapping onto the

quenched Edwards-Wilkinson model (q-EW) via functional RG [14, 16] and aligning with assumption 1b in Section 2.2. Indeed, an alternative way of fixing those dimensions is to impose this value of $\beta$, already known from the q-EW mapping [14]; hyperuniformity with $\varsigma = 2\epsilon/9$ then follows. (A third route to this same answer would be to make an *ansatz* of an emergent rapidity reversal symmetry [44].)

## 5.2   The Physics of Hyperuniformity and the Role of Dangerously Irrelevant Noise

The near-perfect cancellation of active and passive density fluctuations on approach to criticality is remarkable, since the mean density of active particles itself vanishes at the critical point. One may ask: how can fluctuations among a vanishingly small density of active particles perfectly cancel those of a nonvanishing density of passive particles whose fluctuations are either finite (in $d > 4$, where $S_{PP} = p_0$) or even divergent (in $d < 4$, where $S_{PP} \sim q^{-\epsilon/9}$)?

The Gaussian result for $S(q)$ in (19) is again instructive. Here, the ideal-gas-like structure factor (13) for passive particles, $S_{PP} = p_0$, implies that in a cube of linear extent $\lambda$ such that the passive particle number has mean $N_P(\lambda) = p_0 \lambda^d \gg 1$, its standard deviation obeys $\sigma_P(\lambda)^2 \sim p_0 \lambda^d$. To cancel the (Gaussian) fluctuations in passive density requires active particles to have the same standard deviation $\sigma_A(\lambda) = \sigma_P(\lambda)$, but now with a mean of only $N_A(\lambda) = a_0 \lambda^d$. This can be done, with near-Gaussian fluctuations and without creating negative $\rho_A$ locally, only if $\sigma_A(\lambda) \lesssim N_A(\lambda)$. This requires $p_0^{1/2} \lambda^{d/2} \lesssim a_0 \lambda^d$, where $a_0^{-1} \sim \xi^2$, and hence $\lambda \gtrsim \xi^{4/d}$. (Both lengths are here measured in microscopic units.) Hence near-Gaussian fluctuations of the minority active particles *can* cancel the majority passive fluctuations at scales $\lambda \gtrsim \xi$ (which is where hyperuniformity sets in at Gaussian level, see (19)), but only if $d > 4$. In lower dimensions this is not possible. This gives new insight into C-DP's upper critical dimension, $d_c = 4$.

In $d < 4$, the mean number of active particles in a box of size $\lambda$ now varies as $N_A(\lambda) \sim \lambda^d \rho_A \sim \lambda^d \xi^{-\beta/\nu_\perp} \sim \lambda^{4-\epsilon}$ whereas the variance must obey $\sigma_A(\lambda)^2 = \sigma_P(\lambda)^2 \sim S_{PP}(\lambda^{-1})\lambda^d$, giving $\sigma_A(\lambda) \sim \lambda^{2-4\epsilon/9}$. (This estimate follows from the usual relation between compressibility and structure factor, now applied to a subsystem of size $q^{-1} = \lambda$.) Requiring $N_A(\lambda) \gtrsim \sigma_A(\lambda)$ as before yields $\lambda^{2-5\epsilon/9} \gtrsim \xi^{2-5\epsilon/9}$ and hence $\lambda \gtrsim \xi$. This marginal outcome can be extended beyond order $\epsilon$ by use of the scaling relation $2 - \eta_\perp - d = -2\beta/\nu_\perp$ [2]; the details of this calculation can be found in Appendix F. It confirms that in $d < 4$ significantly non-Gaussian fluctuations of $\rho_A$ are needed to avoid negative values: the standard deviation in active particle number is of the same order as its mean in a correlation-length sized box. Without proving that the argument extends across a cascade of shorter scales, as our RG results say it must, we think this makes 'hyperuniformity by cancellation' less mysterious.

A second striking feature of our central result for the RO fixed point, $\varsigma = 2\epsilon/9$, is that it differs from a previous analytical prediction for the hyperuniformity exponent of the C-DP class that arises from the q-EW mapping via a scaling argument [17]. Noting that (as confirmed by our calculations) RO and C-DP share common values of the remaining critical exponents $\beta, \nu_\perp, z$, we ascribe this difference to the fact that the q-EW mapping omits the diffusive noise term in (the nonlinear version of) (26). This is in line with our assumption 1c in Section 2.2.

As already seen in Section 3.3, at Gaussian level if the diffusive noise is suppressed, the entire active phase becomes hyperuniform with $\varsigma = 2$ replacing $\varsigma = 0$. Although analytically proven only in the Gaussian case, hyperuniformity is also found numerically, in the active phase away from the critical region, for several fully nonlinear models with similarly suppressed diffusive noise [26]. We therefore believe the diffusive conservative noise, although often deemed irrelevant in the response-field formalism (where it is treated separately from deterministic diffusion), also alters the correlation function exponent. A term in the action (or field-theoretic operator) that does both these two things is called *dangerously irrelevant* [23].

A dangerously irrelevant operator typically (a) breaks some symmetry and (b) cannot be neglected from the calculation of certain observables, for instance because these contain inverse powers of the irrelevant coupling. The classic example in equilibrium statistical field theory is the Heisenberg model, where the correlation functions perpendicular to the direction of ordering diverge as $q^{-2}$ without cubic anisotropy $v$, but remains finite (of order $1/v$) with it [23]. Thus the fact that $v$ scales to zero at the Heisenberg fixed point does not mean that the correlator can be found without it. Only if $v$ is strictly zero from the outset, so that rotational symmetry is never broken, can anisotropy be ignored. A more recent example in non-equilibrium systems is when a RG-irrelevant activity changes the scaling behaviour of the density correlator in a two-dimensional active nematic system on a frictional substrate [46]. In both cases, an operator seemingly irrelevant via dimension counting affects the correlation function behaviour, while breaking a conservation law/symmetry.

Similarly, the diffusive noise in RO breaks the 'mass-moment' **p**-conservation law introduced in Section 3.3, and as shown there alters the hyperuniformity exponent throughout the active phase of the Gaussian theory. The presence or absence of the same conservation law also affects hyperuniformity exponents in phase-separated active fluids [40], which are governed by a low-temperature fixed point. Hence it is to be expected that the critical RO fixed point (non-conserved **p**) likewise differs from the C-DP one (conserved **p**) via the action of the dangerously irrelevant diffusive noise. If so, the diffusive noise effectively splits the C-DP/RO/q-EW universality class into two. In one sub-class, the diffusive noise is strictly never present (in contrast to scaled to zero under RG flow), so that **p**-conservation is never broken, and the hyperuniformity exponent is $\varsigma = 0 + \epsilon/3$ [17]. In the second sub-class, describing RO as defined in Section 2.2, assumption 1c, there is no conservation law despite the formal irrelevance of the diffusive noise term, and our more singular hyperuniformity exponent $\varsigma = 0 + 2\epsilon/9$ instead prevails.

Notably, the **p**-conserving hyperuniformity exponent $\varsigma = \epsilon/3$ [17], and our **p**-nonconserving one for RO, $\varsigma = 2\epsilon/9$ correspond to two fundamentally different physical pictures of how the fluctuations behave. In the conserved case of [17], the hyperuniformity exponent is predicted from the scaling dimension of the fluctuating $\rho$-field $\rho \sim \nabla^2 \rho_A$. Note also that a 'statistical-tilt' symmetry ensures that the scaling dimensions obey $[\rho - \rho_c] = [\delta\rho]$ to all orders [16, 47]. The latter result is the basis of a previous presentation of the same scaling argument [48] which demands that the density fluctuations are just enough to make some regions drop below $\rho_c$ and become passive (or, on the passive side of the transition, demands that the fluctuations are almost enough to create active regions).

Our hyperuniformity prediction is more singular, and hence implies that density fluctuations are infinitely larger at criticality than the above reasoning would suggest. This looks paradoxical because it would mean that in a system that is very nearly critical but actually passive – which develops hyperuniformity up to very large scales before ceasing finally to move – the density is above the activity threshold in many spatial regions, in which case how can it be passive overall? This paradox is however resolved by our finding that hyperuniformity emerges by *cancellation of active and passive densities*. Accordingly, whatever the overall density fluctuations are, those of the passive particles alone are at least as big. Thus a fluctuation in which $\rho$ rises above $\rho_c$ in some region does *not* imply that active particles are actually present there.

Our cancellation argument is most easily understood in the active phase, for which we have shown above that the active density fluctuations are large enough to compensate the non-hyperuniform fluctuations of passive particles. But this should still work when considering the passive phase, because in the critical region this spends an extremely long time reducing the low-$q$ density fluctuations towards hyperuniformity while there are still active particles, whose density then finally itself decays to zero. At least on the active side, it is known from

simulations that close to criticality large regions of the system are purely passive, and remain so for long periods until a distant patch of activity diffuses into that region [4,49]. This is only marginally possible with $\varsigma = \epsilon/3$ (since $\rho - \rho_c \sim \delta\rho$), but easy to explain with our exponent $\varsigma = 2\epsilon/9$ (less hyperuniform than the former).

We believe the above arguments to expose an interesting physical distinction between the kinds of density fluctuations arising with and without **p**-conservation, that merits further study by use of microscopic models. However it is not clear at this stage whether such models can easily generate reliable data to distinguish hyperuniformity exponents at the critical point. In particular, existing simulations of **p**-conserving models, while showing hyperuniformity throughout the active phase, have often been interpreted using the same critical exponent (at the absorbing state transition itself) as their non-conserving counterparts [8,9,26]. However this might in part stem from the absence until now of any clear theoretical argument to the contrary.

Finally, in the Table below we compare our hyperuniformity exponent for RO calculated to $O(\epsilon)$, with the result found C-DP/Manna (in the absence of conservative noise) [17], calculated to $O(\epsilon^3)$, and with numerical simulations. Subsequent to [8, 9, 26], a numerical work [50] compared critical exponents for RO and a "biased RO" (BRO) model that observes **p**-conservation. There it was found that in three dimensions, while the two models exhibit identical critical exponent $\beta$, their hyperuniformity exponents differ somewhat: $\varsigma_{\text{RO}} = 0.24 \pm 0.02$ and $\varsigma_{\text{BRO}} = 0.26 \pm 0.02$. This numerical discrepancy is suggestive but inconclusive; we look forward to future numerical simulations that further elucidate the exponents involved.

| Dimension | RO/**p**-non-conserving | C-DP/Manna/**p**-conserving |
|---|---|---|
| $d = 3$ | 0.22 | 0.29 {0.33} [17] |
| $d = 2$ | 0.44 | 0.49 {0.66} [17] |
| $d = 3$ (Numerical) | $0.24 \pm 0.02$ [50] | $0.26 \pm 0.02$ [50] |
| $d = 2$ (Numerical) | $0.45 \pm 0.03$ [8,9] | $\approx 0.45$ [26,50] |

Table 1: Hyperuniformity exponent comparison in $d = 3, 2$, found to order $\epsilon$ (this work, without **p**-conservation) for RO, and to order $\epsilon^3$ via the q-EW mapping of [17] for C-DP/Manna (with **p**-conservation). In the latter, the one-loop prediction to $O(\epsilon)$ also shown as $\{\cdot\}$. Also included are published numerical results for $\varsigma$ with and without **p**-conservation.

# 6 Concluding Remarks

The hyperuniformity exponent $\varsigma$ describes a signature feature of the Random Organization (RO) universality class, manifested in the physics of the absorbing state transition for dilute colloids under periodic shearing [3–5], in similar transitions at high density in colloids and granular media [51–53], and in other reaction-diffusion processes with many absorbing states [2]. Our calculation of this exponent, $\varsigma = 2\epsilon/9$ to order $\epsilon = 4 - d$, has shed light on many aspects of RO physics, including the following:

(i) A form of hyperuniformity is present even in the Gaussian limit of the theory, which prevails in $d > d_c = 4$. Study of this theory shows in detail how hyperuniformity in RO emerges via near-perfect anticorrelation of active and passive densities that are not separately hyperuniform but have finite (for $d > 4$) or divergent (for $d < 4$) fluctuations.

(ii) The RG calculations that we perform at one-loop (order $\epsilon$) expose a range of technical difficulties that have previously hindered development of the perturbative RG for absorbing state transitions of this type. While we have not overcome all of these difficulties from first

principles, they can be gathered into a small number of assumptions (laid out in Section 2.2), such as overall renormalizability; a pattern of cancellations among terms akin to that in the 3D tricritical Ising model; and hyperuniformity rather than divergence of the total density fluctuations.

(iii) Diffusive noise splits the C-DP/RO/q-EW universality class into two subclasses by breaking a conservation law on the centre of mass of the total particle density. The RO subclass, in which the conservation law is broken by the diffusive noise (which is dangerously irrelevant) gives a more singular hyperuniformity exponent than one found via a mapping to q-EW (quenched Edwards-Wilkinson) in which the conservation law is sustained. This has implications for the physics of hyperuniformity which merit further study using microscopic models with and without the conservation law [8, 9, 26].

(iv) Our successful completion (albeit subject to the stated assumptions) of a relatively traditional perturbative RG scheme at one-loop level is itself a surprising achievement, since the treatment of other models within the C-DP/RO/q-EW class have required the use of functional RG [16]. The need for functional methods has been ascribed to the presence of infinitely many relevant operators. We do not encounter these in our Doi-Peliti representation, yet our order $\epsilon$ results coincide with the functional RG results for the three exponents $(\beta, \nu_\perp, z)$ that are common to the two subclasses mentioned in (iii) above. (We do not expect the same $\varsigma$ for the reasons stated there.) This suggests that the various mappings between different models and field theories that make up the wider C-DP/RO/q-EW universality class may not be completely understood, and in particular that an infinite number of operators in one case may become a finite number in another, at least for the purposes of one-loop calculations. If so, this suggests Doi-Peliti theory might be more widely useful than currently supposed, potentially as a result of retaining particle entities [28].

We believe these findings to be significant advances towards a more complete understanding of RO physics, and hope they will drive further numerical and experimental investigations of this important class of problems.

## Acknowledgements

We thank Marius Bothe, Cathelijne ter Burg, Rob Jack, Dov Levine, Cesare Nardini, Kay Wiese and Frederic van Wijland for helpful discussions. Work funded in part by the European Research Council under the Horizon 2020 Programme, ERC Grant Agreement No. 740269. XM thanks the Cambridge Commonwealth, European and International Trust, China Scholarship Council, and Trinity College Cambridge for a joint studentship. JP was supported through a UKRI Future Leaders Fellowship (MR/T018429/1 to Philipp Thomas).

## A  Fixed-Point Manifold and exploiting Redundant Relevant Variables

In this Appendix, we lay out the details underlying assumption 1d in Section 2.2.

In many dynamical field theories such as the $\phi^4$ theory, individual coupling constants in the bare theory are also the effective coupling constants of the renormalised one. This means that the RG fixed points are determined by where these coupling constants are scale-invariant under renormalization. Therefore, one can define a 'fixed point action' in which all the relevant coupling constants take their fixed point values. It is best practice to include all possible couplings in the bare theory, so as to ensure that the starting point action is within $\mathcal{O}(\epsilon)$ of the unique fixed point action. (In particular, if a coupling is omitted that is not $\mathcal{O}(\epsilon)$ but $\mathcal{O}(1)$ at

the fixed point, then even the Gaussian level, $\mathcal{O}(1)$ theory will be incorrect.)

In the RO theory constructed from the Doi-Peliti formalism, however, there is no longer a unique fixed point action, but a *manifold* of fixed point actions (the FPM). This is because each microscopic reaction process, under performing second quantisation and the Doi-shift, is separated into several coupling constants; products, rather than individuals, among these different coupling constants are then used to construct loops when doing the RG scheme (see derivation in Section 3). Importantly, since each reaction produces more coupling constants than there are independent effective coupling constants, there arise degeneracies of the conventional 'fixed point action'. These degeneracies define the fixed point manifold. (Note that it is not just a manifold whose elements map onto each other under RG, but a manifold of fixed points, each of which maps onto itself.) As discussed at the Gaussian level in Section 3.2, any of these fixed points is a valid representative of the universality class and will therefore have the same critical exponents as any other.

As one decreases the dimension to below four, the Gaussian fixed point manifold is no longer stable. We argue that when we perform the RG for $d = 4 - \epsilon$ dimensions, we first start somewhere on the Gaussian FPM and add a subset of nonlinearities indicated from our microscopic RO theory, which gives an appropriate unstable direction to leave the Gaussian fixed point manifold. Any subset must effectively capture **all** relevant nonlinearities at a certain critical fixed point on the $d = 4 - \epsilon$ fixed point manifold, which we assume to be $\mathcal{O}(\epsilon)$ away from the Gaussian FPM. Our subset of nonlinearities comes from the minimal set of reaction and diffusion processes within the RO universality class that do not promote 'problematic' couplings (as discussed in Appendix B); other potential forms of the fixed point action, while still on the $d = 4 - \epsilon$ FPM, contain these problematic couplings which induce algebraic (not logarithmic) IR/UV divergences. A similar employment of degeneracies caused by redundant parameters is illustrated in [41].

## B  Problematic Divergences: Comparison to Tricritical Ising

In the tricritical Ising model, there exist algebraic (not logarithmic) IR divergences $\sim 1/m$ or $\sim \ln m/m$ at the upper critical dimension $d_c = 3$. These problematic diagrams cannot be simply eliminated individually, and are precisely those involving a four-point vertex that is generated under RG even if absent at Gaussian level. However, as one should anticipate, these divergences must cancel collectively in order to give a renormalisable tricritical Ising theory near three dimensions [35], since any remaining quartic term would immediately raise $d_c$ from three to four (and recover the standard Wilson-Fisher critical point, not the tricritical one).

Our RG analysis for RO shows some very similar features to the tricritical Ising model, especially in the IR regime. While we are interested in the RO fixed point which has an upper critical dimension of four, there exist unstable directions to a general theory (such as dynamical percolation [15], or pair contact process [34]) which has an upper critical dimension of six. Also, like in the tricritical Ising, there exist couplings such as , whose inclusion introduces loop integrals that have non-renormalisable divergences in the form of negative powers of $m$. An example of such a 'problematic' term is as follows:

$$\text{} = \frac{1}{\epsilon_p} \int \frac{1}{q^2 + \epsilon_p} d^d q \rightarrow \left( \frac{\Lambda^{d-2}}{\epsilon_p} \right) \text{for dimension } d > 2 \qquad (B.1)$$

Like the tricritical Ising model, the appearance of non-renormalisable IR divergences is likely a result of the RO theory lying in the submanifold of a generic theory with a higher

| | $u_4$ | $\alpha_3$ |
|---|---|---|
| Theory | Tricritical in Ising ($d_c = 3$ vs 4) | RO in percolation ($d_c = 4$ vs 6) |
| Generated? | Yes | Yes |
| Divergence | $\sim u_4 \cdot 1/m$ | $\sim \Lambda^{d-2}/\epsilon_p$ |
| Engineering dimension | $u_4(\zeta) = u_4\zeta$ (relevant) | Marginal |

Table 2: Comparison between four-point vertex in tricritical ising and 'problematic' couplings in RO. Here $m, \epsilon_p$ are the mass term for tricritical Ising and RO respectively.

upper critical dimension. However, the additional non-renormalisable UV divergence observed in RO indicates a more serious challenge that we have not so far been able to resolve from first principles. Therefore, in Section 2, we have phrased this comparison as an **assumption** (2c) that a similar cancellation across super-divergent terms should occur in RO, leading to suppressed $\alpha_3, \sigma_{5,6}$. Evidence for this assumption is that, by making it, we can not only sustain $d_c = 4$ as is known from the q-EW mapping of [13–15], but also reproduce the three standard exponents $\beta, \nu_\perp, z$ of the C-DP/RO/q-EW class found via that mapping [16].

## C  Effective Couplings at the Gaussian Level

In Section 4.3.3, we argued that to preserve symmetries between coupling constants while suppressing the vertices (31) that induce algebraic (not logarithmic) UV divergences, we consider corrections to $\alpha_1$ and $\alpha_2$; the $Z$-factor for the latter contains corrections to $\alpha_3$. In this Appendix, we show that this is in accordance with what is observed in the Gaussian structure factors. At Gaussian level, tree-level diagrams for the two-point correlation functions produce

$$\text{Cov}[A(q,t)A(-q,t)] = \alpha_1 \left[ \frac{1}{Dq^2 + \tau_p} \right] + a_0 \tag{C.1}$$

$$\text{Cov}[A(q,t)P(-q,t)] + \text{Cov}[A(-q,t)P(q,t)] = -2\alpha_1 \left[ \frac{1}{(Dq^2 + \tau_p)} \right] \tag{C.2}$$

$$\text{Cov}[P(q,t)P(-q,t)] = \alpha_3 \left[ \frac{1}{\tau_p} \right] + p_0 \tag{C.3}$$

$$\text{Cov}[\rho(q,t)\rho(-q,t)] = a_0 + p_0 - \frac{\alpha_1}{Dq^2 + \tau_p} + \frac{\alpha_3}{\tau_p} \tag{C.4}$$

As stated in assumptions 2a, 2c in Section 2.2, the vertex $\alpha_3$ needs to be suppressed for a renormalisable theory both for $d > d_c = 4$ and $d < d_c = 4$. Upon suppression of $\alpha_3$, the two-point correlation functions are solely dependent on $\alpha_1 = -(\alpha_2 + \alpha_3)$. That the theory in $d < 4$ is dependent only on one parameter, namely $\alpha_1 = -\alpha_2$, once $\alpha_3$ is assumed to vanish, provides some level of confidence for the argument.

## D  Evaluation of all Loop Corrections

The UV-divergent part of the diagrams required to determine the exponents in this work are listed in Tables 3 and 4 below. These are to be used according to the following guide:

- The first row shows the diagrammatic representations of the loops

- The second row gives the loop integral, calculated using dimensional regularization as shown for example in Eqs. (40) and (44).

- The rows labelled as 'prefactors' (e.g. $\alpha_2$ prefactors in the third row) provide the couplings whose vertices are placed at the black dots in the loop diagrams of the first row. As an example, for Loop A to correct $\alpha_2$, the couplings required are $\lambda_2, \sigma_3, \alpha_1$.

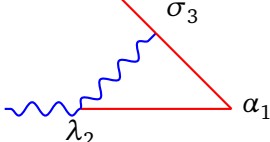

- In the same rows, an underlined 2 is the symmetry factor for Feynman diagrams, which arises if and only if one of $\sigma_1$, $\sigma_2$ or $\alpha_1$ appears in the prefactor. This is because $\sigma_1$, $\sigma_2$ and $\alpha_1$ have two outgoing active propagators which can be interchanged leading to a symmetry factor of 2.

- The second loop in each table requires caution as highlighted in Section 4.3.3. For simplicity, we have used the prefactors for the symmetric counterparts $\alpha_1$, $\sigma_1$ and $\sigma_2$, with an additional negative sign.

- The rows labelled as 'flow function' (e.g. $\alpha_2$, $\alpha_1$ flow function in the fourth row) shows what appears in the $Z$-factor of the corresponding couplings. Each entry in the table is a constant multiple of one of the three effective coupling constants. As an example, the '$u_R$' for the first loop in the row of '$\alpha_2$, $\alpha_1$ flow function' is calculated by $\frac{1}{2} \frac{(\epsilon_p)^{d/2-2}}{D^{d/2}} \frac{A_d}{(4-d)} \cdot 2\lambda_2\sigma_3\alpha_1/\alpha_2 = -\lambda_2\sigma_3 \frac{(\epsilon_p)^{d/2-2}}{D^{d/2}} \frac{A_d}{(4-d)} = u_R$.

| | | | | | |
|---|---|---|---|---|---|
| Loop Integral | $\frac{1}{2}\frac{(\epsilon_p)^{d/2-2}}{D^{d/2}}\frac{A_d}{(4-d)}$ | $\frac{(\epsilon_p)^{d/2-2}}{D^{d/2}}\frac{A_d}{(4-d)}$ | $\frac{(\epsilon_p)^{d/2-2}}{D^{d/2}}\frac{A_d}{(4-d)}$ | $\frac{1}{2}\frac{(\epsilon_p)^{d/2-2}}{D^{d/2}}\frac{A_d}{(4-d)}$ | $\frac{1}{2}\frac{(\epsilon_p)^{d/2-2}}{D^{d/2}}\frac{A_d}{(4-d)}$ |
| $\alpha_2$ prefactors | $\underline{2}\lambda_2\sigma_3\alpha_1$ | $-\underline{2}\lambda_1\sigma_1\alpha_2$ | $\lambda_2\alpha_2\sigma_4$ | $\lambda_2\sigma_3\alpha_2$ | $\lambda_2\sigma_3\alpha_2$ |
| $\alpha_2$, $\alpha_1$ flow function | $u_R$ | $2u_R$ | $v_R$ | $-\frac{1}{2}u_R$ | $-\frac{1}{2}u_R$ |
| $\sigma_3$ prefactors | $\underline{2}\lambda_2\sigma_3\sigma_1$ | $-\underline{2}\lambda_1\sigma_1\sigma_3$ | $\lambda_2\sigma_3\sigma_4$ | $\lambda_2\sigma_3\sigma_3$ | $\lambda_2\sigma_3\sigma_3$ |
| $\sigma_3$, $\sigma_1$ flow function | $u_R$ | $2u_R$ | $v_R$ | $-\frac{1}{2}u_R$ | $-\frac{1}{2}u_R$ |
| $\sigma_4$ prefactors | $\underline{2}\lambda_2\sigma_3\sigma_2$ | $-\underline{2}\lambda_1\sigma_1\sigma_4$ | $\lambda_2\sigma_4\sigma_4$ | $\lambda_2\sigma_3\sigma_4$ | $\lambda_2\sigma_3\sigma_4$ |
| $\sigma_4$, $\sigma_2$ flow function | $u_R$ | $2u_R$ | $v_R$ | $-\frac{1}{2}u_R$ | $-\frac{1}{2}u_R$ |
| $\lambda_1$ prefactors | $\underline{2}\lambda_1\lambda_2\sigma_1$ | $\lambda_1\lambda_1\sigma_3$ | $\lambda_1\lambda_2\sigma_4$ | $\lambda_1\lambda_2\sigma_3$ | $\lambda_1\lambda_2\sigma_3$ |
| $\lambda_1$, $\lambda_2$ flow function | $u_R$ | $u_R$ | $v_R$ | $-\frac{1}{2}u_R$ | $-\frac{1}{2}u_R$ |

Table 3: Full list of $\lambda\sigma$-type loops that give non-zero contributions.

Next we provide an example of how to use the tables. We take $\lambda_1$ as an example. Reading from the prefactor row of the tables, the vertex function corresponding to $\lambda_1$, denoted as $\Gamma_{\lambda_1}$, is corrected as (second to last row in each table):

$$\Gamma_{\lambda_1} = \lambda_1 + 2\lambda_1\lambda_2\sigma_1 \frac{1}{2} \frac{(\epsilon_p)^{d/2-2}}{D^{d/2}} \frac{A_d}{4-d} + \lambda_1\lambda_1\sigma_3 \frac{(\epsilon_p)^{d/2-2}}{D^{d/2}} \frac{A_d}{4-d} + \lambda_1\lambda_2\sigma_4 \frac{(\epsilon_p)^{d/2-2}}{D^{d/2}} \frac{A_d}{4-d}$$
$$+ \frac{1}{2}\lambda_1\lambda_2\sigma_3 \frac{(\epsilon_p)^{d/2-2}}{D^{d/2}} \frac{A_d}{4-d} + \frac{1}{2}\lambda_1\lambda_2\sigma_3 \frac{(\epsilon_p)^{d/2-2}}{D^{d/2}} \frac{A_d}{4-d}$$
$$\text{(D.1)}$$

The three terms from the second table cancel each other because $\lambda_1 = -\lambda_2$ (note the different factors of $\frac{1}{2}$ and $\frac{1}{4}$ in the 2nd row which shows the loop integrals).

| | | | |
|---|---|---|---|
| |  |  |  |
| Loop Integral | $\frac{1}{4}\frac{(\epsilon_p)^{d/2-2}}{D^{d/2}}\frac{A_d}{(4-d)}$ | $\frac{1}{2}\frac{(\epsilon_p)^{d/2-2}}{D^{d/2}}\frac{A_d}{(4-d)}$ | $\frac{1}{4}\frac{(\epsilon_p)^{d/2-2}}{D^{d/2}}\frac{A_d}{(4-d)}$ |
| $\sigma_4$ prefactor | $\lambda_2^2\alpha_2\sigma_4$ | $-2\lambda_1\lambda_2\sigma_2\alpha_2$ | $\lambda_2^2\alpha_2\sigma_4$ |
| $\sigma_4,\sigma_2$ flow function | $\frac{1}{4}w_R$ | $-w_R$ | $\frac{1}{4}w_R$ |
| $\lambda_1$ prefactor | $\lambda_1\lambda_2^2\alpha_2$ | $\lambda_1^2\lambda_2\alpha_2$ | $\lambda_1\lambda_2^2\alpha_2$ |
| $\lambda_1,\lambda_2$ flow function | $\frac{1}{4}w_R$ | $-\frac{1}{2}w_R$ | $\frac{1}{4}w_R$ |

Table 4: Full list of $\lambda\lambda\alpha$-type loops that give non-zero or non-cancelling contributions.

Extracting out the factor $\lambda_1$ gives the $Z$-factor,

$$
\begin{aligned}
Z_{\lambda_1}Z_{\tilde{a}}^{1/2}Z_{\breve{a}}^{1/2}Z_{\check{p}}^{1/2} &= 1 + \lambda_2\sigma_1\frac{(\epsilon_p)^{d/2-2}}{D^{d/2}}\frac{A_d}{4-d} + \lambda_1\sigma_3\frac{(\epsilon_p)^{d/2-2}}{D^{d/2}}\frac{A_d}{4-d} + \lambda_2\sigma_4\frac{(\epsilon_p)^{d/2-2}}{D^{d/2}}\frac{A_d}{4-d} \\
&\quad + \frac{1}{2}\lambda_2\sigma_3\frac{(\epsilon_p)^{d/2-2}}{D^{d/2}}\frac{A_d}{4-d} + \frac{1}{2}\lambda_2\sigma_3\frac{(\epsilon_p)^{d/2-2}}{D^{d/2}}\frac{A_d}{4-d} \\
&= 1 + \frac{\lambda_1\sigma_3}{D^{d/2}}\frac{A_d\zeta^{-\epsilon}}{\epsilon} + \frac{\lambda_2\sigma_4}{D^{d/2}}\frac{A_d\zeta^{-\epsilon}}{\epsilon}
\end{aligned}
\tag{D.2}
$$

The flow function is hence

$$
\gamma_{\lambda_1} := \zeta\frac{\partial}{\partial\zeta}\bigg|_0 \ln Z_{\lambda_1} = -2 + \frac{d}{2} - u_R - v_R - \frac{1}{2}(\gamma_{\tilde{a}} + \gamma_{\breve{a}} + \gamma_{\check{p}})
\tag{D.3}
$$

Alternatively, the 'flow function' rows of the tables are constructed so that the $Z$-factors can be read off easily. Each entry is a constant multiple of an effective coupling constant: the constant multiple is calculated by multiplying the loop integral with the symmetry factors, and the effective coupling constant is identified by looking at which interaction vertices are used. Again using $\lambda_1$ as an example, its flow function can be read from the tables immediately (last row in each table),

$$
\begin{aligned}
\gamma_{\lambda_1} &= -2 + \frac{d}{2} - (u_R + u_R + v_R - \frac{1}{2}u_R - \frac{1}{2}u_R + \frac{1}{4}w_R - \frac{1}{2}w_R + \frac{1}{4}w_R) - \frac{1}{2}(\gamma_{\tilde{a}} + \gamma_{\breve{a}} + \gamma_{\check{p}}) \\
&= -2 + \frac{d}{2} - u_R - v_R - \frac{1}{2}(\gamma_{\tilde{a}} + \gamma_{\breve{a}} + \gamma_{\check{p}})
\end{aligned}
\tag{D.4}
$$

As above for the vertex function, the three terms of the 2nd table cancelled each other.

Similarly, flow functions for other vertices are,

$$
\gamma_{\sigma_3} = -2 + \frac{d}{2} - 2u_R - v_R - \frac{1}{2}(\gamma_{\tilde{a}} + \gamma_{\check{p}} + \gamma_{\breve{a}})
\tag{D.5}
$$

$$
\gamma_{\sigma_4} = -2 + \frac{d}{2} - 2u_R - v_R + \frac{w_R}{2} - \frac{1}{2}(\gamma_{\tilde{a}} + \gamma_{\check{p}} + \gamma_{\check{p}})
\tag{D.6}
$$

$$
\gamma_{\alpha_2} = -2 - 2u_R - v_R - \frac{1}{2}(\gamma_{\tilde{a}} + \gamma_{\check{p}})
\tag{D.7}
$$

Therefore the renormalisation group beta functions for the effective coupling constants

read

$$\beta_u = \zeta \frac{\partial}{\partial \zeta}\bigg|_0 u_R = u_R(\gamma_{\lambda_1} + \gamma_{\sigma_3} - \frac{d}{2}\gamma_D) = u_R\left[-\epsilon - 3u_R - 3v_R - \frac{1}{2}w_R + \mathcal{O}(\epsilon^2)\right] \tag{D.8}$$

$$\beta_v = \zeta \frac{\partial}{\partial \zeta}\bigg|_0 v_R = v_R(\gamma_{\lambda_2} + \gamma_{\sigma_4} - \frac{d}{2}\gamma_D) = v_R\left[-\epsilon - 2u_R - 4v_R - \frac{1}{2}w_R + \mathcal{O}(\epsilon^2)\right] \tag{D.9}$$

$$\beta_w = \zeta \frac{\partial}{\partial \zeta}\bigg|_0 w_R = w_R(2\gamma_{\lambda_2} + \gamma_{\alpha_2} - \gamma_{\epsilon_p} - \frac{d}{2}\gamma_D) = w_R\left[-\epsilon - 4u_R - 5v_R - \frac{3}{2}w_R + \mathcal{O}(\epsilon^2)\right] \tag{D.10}$$

# E  List of One-loop Diagrams

Alongside the contributing loops shown in Tables 3 and 4 in Appendix D above, we now show all remaining (non-contributing or cancelling) loops at 1-loop order, for completeness.

(i) The following loops are 'non-contributing', meaning that the loop integral does not show a divergence close to the upper critical dimension, and therefore only gives subdominant corrections.

Non-contributing $\lambda\sigma$-type loops:

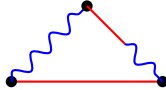

Non-contributing $\lambda\lambda\alpha$-type loops:

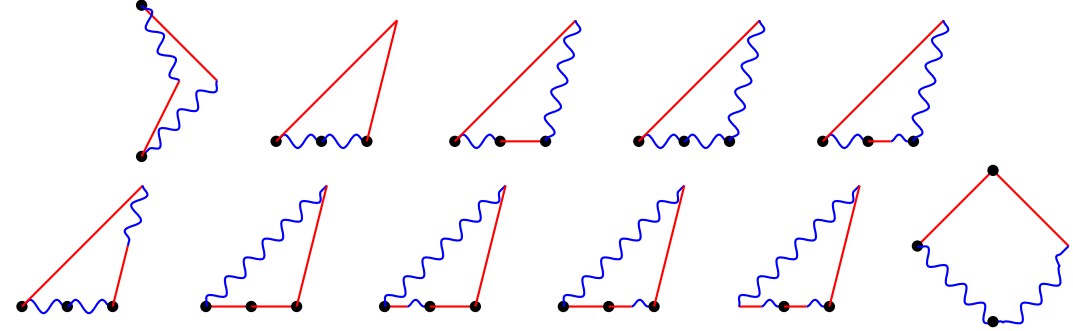

(ii) We now turn to 'cancelling' loops, meaning that although their contribution is of the correct order, they would eventually cancel with each other in vertex corrections. Evaluation of the loop integral gives the same value within each of the pairs shown below, and the prefactors in which these enter any given vertex correction have a ratio of $\lambda_1 : \lambda_2$. By the symmetries (45) established via particle number conservation, $\lambda_1 + \lambda_2 \equiv 0$; as a direct result of this symmetry, the below loop corrections grouped in brackets cancel:

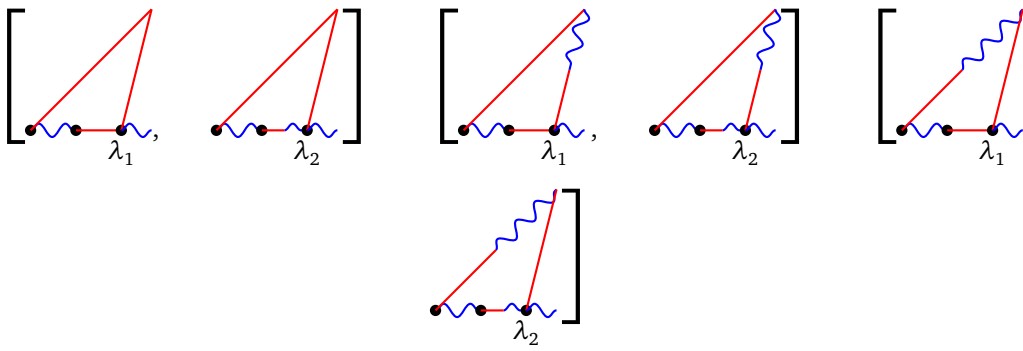

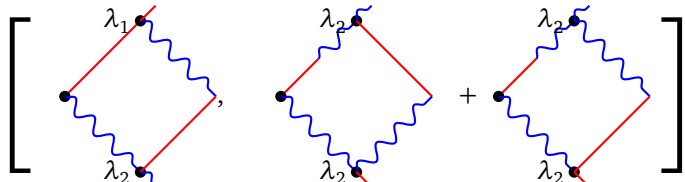

## F    Cancellation of Density Fluctuations beyond $O(\epsilon)$

In $d < 4$, the mean number of active particles in a box of size $\lambda$ is given by the active density multiplied by the size of the box, $N_A(\lambda) \sim \lambda^d \rho_A \sim \lambda^d \xi^{-\beta/\nu_\perp}$. For a system of size $\lambda = q^{-1}$, to cancel the finite variance from passive density, the active density variance $\sigma_A(\lambda)^2 = \sigma_P(\lambda)^2 \sim S_{PP}(\lambda^{-1})\lambda^d \sim \lambda^{2-\eta_\perp}$, where $\eta_\perp$ is defined via [2]

$$S_{PP}(q) \sim S_{AA}(q) \sim q^{-2+\eta_\perp}. \tag{F.1}$$

Now to avoid passive densities, we require $N_A(\lambda) \gtrsim \sigma_A(\lambda)$, which gives

$$\lambda^d \xi^{-\beta/\nu_\perp} \gtrsim \lambda^{\frac{2-\eta_\perp+d}{2}} \Rightarrow \lambda^{d+\eta_\perp-2} \gtrsim \xi^{2\beta/\nu_\perp}. \tag{F.2}$$

Using the scaling relation [2]

$$\eta_\perp + d - 2 = 2\beta/\nu_\perp, \tag{F.3}$$

we obtain $\lambda \gtrsim \xi$ to all orders of $\epsilon$.

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
