# Peer review of "Hyperuniformity at the Absorbing State Transition: Perturbative RG for Random Organization"

_SciPost Physics_

## Round 1 · Referee Report · Anonymous (Referee 1) · 2025-9-29

Strengths

  1. Interesting subject
  2. An interesting proposal for a reaction-diffusion system exhibiting hyperuniformity
  3. A possible explanation for the difference between the model studied and those previously studied by the functional renormalisation group (dangerously irrelevant operator)

Weaknesses

  1. Too many unjustified assertions (some being very problematic)
  2. Too many vague statements/misuse of standard concepts of perturbative renormalization

Report

In their paper: ``Hyperuniformity at the absorbing state transition...'' the authors compute the critical exponents in an $\epsilon=4-d$ expansion of a class of models showing hyperuniformity of the total density of particles. This article deals with a very interesting subject, aims to be clear, and presents results that seem reasonable and that reproduce known results when they should. However, even though the authors strive to explain as clearly as possible the assumptions on which their calculations are based, there remain too many unexplained/unjustified points to recommend the publication of this article as it stands.

I will first list my main comments in order of importance, followed by the less important points.

Remark 1. The authors repeatedly mention the existence of non-renormalizable infrared (IR) divergences in their theory (introduction pages 5 et 6, Appendix B). But the very concept of renormalizable or non-renormalizable IR divergences is meaningless: Perturbative renormalization only deals with the recursive elimination of UV divergences by modification of the short-distance structure of the theory (under the form of the addition of local counter-terms). IR singularities should certainly not be eliminated and in any case it is impossible to do so (at least if we want to remain in the same universality class). These singularities can be regularized by slightly shifting the system away from criticality by modifying a control parameter (temperature, reaction rates, etc.). But the singularities are not eliminated: fine-tuning the control parameter to bring it back to its critical value causes the singularities to reappear. As this is the universally accepted vocabulary of perturbative renormalisation, I suggest that the authors adhere to it.

Remark 1' (less important than 1). In somewhat the same vein, the authors mention several times that certain couplings can be generated at the fixed point. This is not how renormalization group (RG) works. When couplings that are absent in the bare action are generated, they are generically generated from the first RG step, not only at the fixed point (which implies that they are also important when the system is not critical).

Remark 2. I know of two circumstances that prevent a coupling to be generated along the RG flow: (i) symmetries: they exclude non-symmetric terms to be generated under RG transformations and (ii) algebraic constraints such as for instance causality and Ito's prescription which imply that both the bare action and the effective action (the generating functional of 1PI vertex functions) are proportional to the response field (no term without a response field can be generated). Except for these two cases, all possible couplings are generated as soon as coarse-graining, that is, RG transformations, is performed. This implies that if a set of n relevant (in the RG sense) couplings is absent in the RG flow near a fixed point, n fine tunings of coupling constants have been implemented in the bare action. This is what happens for instance at the tricritical fixed point of the Ising model: it is twice unstable in the IR which means that two fine tunings of bare parameters are required to reach the tricritical hypersurface (which is therefore of codimension two in the infinite dimensional coupling constant space). I therefore agree with the authors when they state that a model may have fewer divergent terms than one might naively expect from symmetries and power counting, but I would like to emphasize a crucial point that in my opinion has been overlooked in the article: this comes at a price, namely the fine tuning of as many couplings as there are relevant directions at the fixed point. Thus, the analogy with tricriticality put forward by the authors does not look relevant to me because if this is the mechanism at work in their case, it means that they are not studying a critical phase transition but rather a multicritical one, a situation that does not seem to correspond to the microscopic system under study. This crucial point requires clarification.

Remark 3. Symmetries play a considerable role in the definition of universality classes and the authors are well aware of this. However, they do not perform a systematic study of all the symmetries of the bare action coming from the master Eq.(1), that is, before the Doi shift. The interaction part of the action reads: $\kappa a^\dagger(a^\dagger-p^\dagger)ap +\mu (p^\dagger - a^\dagger)a$. This action has an obvious U(1) symmetry consisting in multiplying the a and p fields by a phase factor exp(i alpha) and the complex conjugate fields by exp(-i alpha) with alpha a real number. The conserved charge associated with this symmetry is the total particle number and its conservation is therefore clearly related to a symmetry of the action. Notice that performing the Doi shift hides this symmetry (for this reason, this shift can be dangerous as emphasized by Cardy and Tauber in their article: ``Field Theory of Branching and Annihilating Random Walks''). Of course, the translation of the annihilation fields by $a_0$ and $p_0$ also hides this symmetry.

Remark 3'. The authors repeatedly refer to ``implicit symmetries'' (in the introduction for instance) without defining what this means and what they consist of (I do not know what an implicit symmetry is).

Remark 4. The authors write in section 3.2 that the shift by $a_0$ and $p_0$ refer to ``the mean densities of a set of Poisson-distributed active and passive particles initialised in the distant past''. This seems weird because:

(i) this contradicts what the authors do when they translate the $a(x,t)$ and $p(x,t)$ fields by their mean field values $a_0=\rho_A=\rho-\mu/\kappa$, $p_0=\mu/\kappa$. These values are independent of the initial distributions of the A and P particles (and they depend on $\mu$ and $\kappa$ that are of course independent of the initial distributions),

(ii) in the long time limit, the initial distributions of the particles play no longer any role (at least for universal quantities) and only the total density remains unchanged.

Remark 4'. Then the authors claim that at gaussian level these mean densities remain unchanged under time evolution (before Eq.(5)). This is not correct if we consider the bilinear terms in the creation and annihilation operators of the A and P particles that describe diffusion and transmutation of the A particles into P particles: Taking only these terms into account, all A particles are transmutated into P particles in the long time limit, so that the average density of A and P particles evolves due to the Gaussian terms.

Remark 5. The authors claim that the presence of a dangerously irrelevant term in their theory plays an important role and modifies the hyperuniformity exponent. Their claim relies on the presence of a diffusive noise in the active phase which is not taken into account in some other version of the model. However this noise term is only explicit in the Cole-Hopf formulation of the model, not in that of Doi-Peliti where it is automatically taken into account. Consequently, the authors cannot explicitly show a term that would be dangerously irrelevant in their perturbative calculation, and the meaning of all this in this formulation is unclear (see also Remark 12 below).

Remark 6. The authors find that except for the hyperuniformity exponent $\zeta$, all other exponents are identical to those found in the other versions of the model (studied by functional RG). As said previously, the authors claim that these other versions of the model neglect the diffusive noise which the authors claim to be dangerously irrelevant. From the above statements, we can conclude that all exponents except $\zeta$ are insensitive to the dangerously irrelevant term and therefore can be computed by ignoring this subtle aspect of the model. Therefore, we can conclude that they can be computed by usual means, that is, coming from the passive phase. Then, at least for the computation of these exponents, why not studying the model directly at criticality, that is, without performing any translation of the a and p fields? This is what is usually done in reaction-diffusion systems (and without performing the Doi shift that hides the symmetry). Doing this makes trivial to prove diagrammatically that the coupling constants in front of the $a^\dagger{}^2 a p$ term and the $-a^\dagger p^\dagger a p$ term remain identical all along the RG flow if they are in the bare action (these two terms are renormalized by the same diagrams). Actually, it seems to me that working with this parameterization of the field theory makes it perturbatively trivial because the four-point 1PI functions corresponding to the two bare terms mentioned above are only renormalized by a chain of bubble diagrams that can probably be resummed. Moreover, the propagator of the $a$ and $p$ fields are not renormalized and neither the transmutation parameter $\mu$. The whole theory looks extremely simple in this parameterization except for the strange propagator of the p field: Only $\kappa$ runs. There may be a subtle problem here that renders the above argument irrelevant, but in that case the authors should explain it in detail.

Remark 7. The authors repeatedly mention the existence of a fixed point manifold but never explain what this manifold consists in. It seems crucial for a good understanding of the paper that they very precisely explain what this manifold is.

And now perhaps less important remaks.

Remark 8. The authors present a power counting argument in Eqs. (32-34) based on the assumption that the a and p fields together with their response fields have the same dimension d/2. This asumption is unjustified. It is well known for non-equilibrium systems that the quadratic terms of the action only determine the scaling dimension of the product (field . response field), which is d, and not of the two fields separately. In the absence of a symmetry exchanging the fields and the response fields, there is no reason for them to have the same scaling dimension. Such symmetry exists for directed percolation, but there are many other systems for which this is not the case. The only systematic method I am aware of for attributing a scaling dimension to the fields of a given model is to perform a one-loop calculation and then to give a scaling dimension to the fields and, as a consequence to the couplings, so as to reproduce the degree of UV divergence of each individual graphs (the superficial degree of divergence of a graph is entirely determined by dimensional analysis once the dimensions of the couplings are known). Once fixed at one loop, it is valid at all loop orders. I see no reason why the power counting given by the authors should be correct.

Note that physical arguments may sometimes require that two interaction terms in the bare action are equally relevant (otherwise, compared to the other, one term could be neglected and the physics would then be completely different). In this case, this gives an additional equation for the field dimensions, which is sufficient to fix them completely, thus avoiding a one-loop calculation (for a striking example, see ‘Pair Contact Process with Diffusion: Failure of Master Equation Field Theory’ by H.-K. Janssen et al.).

Remark 9. As a consequence of Remark 8, the upper critical dimension of the model is not computed by the authors who assume that it is 4. This situation is clearly unsatisfactory and it would be interesting to see whether implementing the program outlined in Remark 6 makes it possible to determine the upper critical dimension (which, in principle, is not affacted by a dangerously irrelevant operator).

Remark 10. The authors claim that the model (after the shifts by $a_0$ and $p_0$) presents UV non-renormalizable divergences that should cancel for the RO universality class broadly following the same scenario as the tricritical Ising model in d=3 (see Appendix B, for instance and section 4.3.3). They claim that in this latter case, the $\phi^4$ coupling is generically generated along the RG flow even it is absent in the bare action and it produces dangerous UV divergences. This statement looks weird to me: The $\phi^4$ term is super-renormalizable in $d<4$ and I doubt that it generates nasty UV divergences. It is true that being super-renormalizable in the UV, it produces strong IR divergences, but this is a different story. Here again, it seems to me that the authors mix UV and IR divergences (see for instance the sentence in B: ``... non-renormalizable divergences in the form of negative powers of m.'')

Remark 11. The authors introduce the notion of effective couplings (bottom of page 11 for instance) without defining them. I do not know what an effective coupling is. Is it a coupling appearing in the effective action?

Remark 12. The authors use the Cole-Hopf transformation in the 3.3 section. However, this transformation is unjustified at the operator level: if it were justified, then the annihilation operator would be invertible which it is certainly not. The Cole-Hopf transformation is heuristically interesting but should be used with great care. I suggest that the authors mention this point.

Remark 13. Eq. (24) cannot be valid if the noise field $\eta$ is real. This is not a detail at least beyond perturbation theory as can be seen in the supplemental material of the article: ``Langevin equations for reaction-diffusion processes'' by Benitez et al. where the deformations of contours are performed by taking care of the reality of the fields.

Remark 14. There are typos in Eqs. (10) (a $d\omega'$) and (23) (a square and a factor of 2).

Remark 15. The simplest dangerously irrelevant operator in equilibrium theory is the $\phi^4$ coupling in $d\ge4$. This term does not break the O(N) symmetry and in general a dangerously irrelevant operator does not need to break any symmetry (although some of them do break symmetries) contrary to what is suggested in section 5.2: ``A dangerously irrelevant operator typically (a) breaks some symmetry...''

Remark 16. In apppendix A, the authors mention a ``fixed point action in which all the relevant coupling constants take their fixed point values''. In my opinion, this sentence is a bit misleading. First, it is the effective action that can be at a fixed point, not a bare one (typically, a bare action involves a finite number of couplings while the effective action involves infinitely many couplings). Second, only dimensionless couplings can reach a fixed point value, not the dimensionful ones. Third, at a fixed point associated with a second order phase transition, there is only one relevant direction (loosely speaking, the mass direction) and all the other ones are irrelevant. They all have a fixed point value, not only the relevant one.

For all the reasons listed above, I believe this article requires significant revisions before it can be published.

Attachment

Recommendation

Ask for major revision

  • validity: low
  • significance: ok
  • originality: good
  • clarity: low
  • formatting: excellent
  • grammar: perfect

Author:  Xiao Ma  on 2025-12-07  [id 6115]

(in reply to Report 1 on 2025-09-29)

Dear Editors and Referee,

We have carefully reviewed all comments and suggestions. The feedback we received was highly valuable and has greatly contributed to strengthening the quality of our manuscript. We have made appropriate revisions to the manuscript to address many of the points raised.

In our reply to the referees' comments in the attached pdf file, our responses are highlighted in purple. Modifications done to the main text are highlighted in blue. The location of these modifications in the text can be identified via the diff file that we have provided alongside page references thereto.

Yours sincerely,

The authors

Attachment:

Response_to_referee_1.pdf

---

## Round 1 · Referee Report · Anonymous (Referee 2) · 2025-10-8

Strengths

see pdf-file

Weaknesses

see pdf-file

Report

see pdf-file

Requested changes

see pdf-file

Attachment

Recommendation

Ask for major revision

  • validity: poor
  • significance: low
  • originality: ok
  • clarity: low
  • formatting: good
  • grammar: excellent

Author:  Xiao Ma  on 2025-12-07  [id 6116]

(in reply to Report 2 on 2025-10-08)

Dear Editors and Referee,

We have carefully reviewed all comments and suggestions. The feedback we received was highly valuable and has greatly contributed to strengthening the quality of our manuscript. We have made appropriate revisions to the manuscript to address many of the points raised.

In our reply to the referees' comments in the attached pdf file, our responses are highlighted in purple. Modifications done to the main text are highlighted in blue. The location of these modifications in the text can be identified via the diff file that we have provided alongside page references thereto.

Yours sincerely,

The authors

Attachment:

Response_to_referee_2.pdf

---

## Editorial Decision

resubmitted